# Traumatic brain injury promotes neurogenesis at the cost of astrogliogenesis in the adult hippocampus of male mice

P. Bielefeld[1,13], A. Martirosyan[2,3,13], S. Martín-Suárez[4,13], A. Apresyan [5,13], G. F. Meerhoff[1], F. Pestana [2,3], S. Poovathingal[2,3], N. Reijner[1], W. Koning[1], R. A. Clement[1], I. Van der Veen[1], E. M. Toledo[1], O. Polzer [1], I. Durá[4], S. Hovhannisyan[6], B. S. Nilges[7,8], A. Bogdoll[7], N. D. Kashikar [7,8], P. J. Lucassen [1], T. G. Belgard [9], J. M. Encinas [4,10,11], M. G. Holt [2,3,12,14] ✉ & C. P. Fitzsimons [1,14] ✉

Traumatic brain injury (TBI) can result in long-lasting changes in hippocampal function. The changes induced by TBI on the hippocampus contribute to cognitive deficits. The adult hippocampus harbors neural stem cells (NSCs) that generate neurons (neurogenesis), and astrocytes (astrogliogenesis). While deregulation of hippocampal NSCs and neurogenesis have been observed after TBI, it is not known how TBI may affect hippocampal astrogliogenesis. Using a controlled cortical impact model of TBI in male mice, single cell RNA sequencing and spatial transcriptomics, we assessed how TBI affected hippocampal NSCs and the neuronal and astroglial lineages derived from them. We observe an increase in NSC-derived neuronal cells and a concomitant decrease in NSC-derived astrocytic cells, together with changes in gene expression and cell dysplasia within the dentate gyrus. Here, we show that TBI modifies NSC fate to promote neurogenesis at the cost of astrogliogenesis and identify specific cell populations as possible targets to counteract TBI-induced cellular changes in the adult hippocampus.

Traumatic brain injury (TBI) is a major global health problem linked to everyday life events such as domestic activities, participation in (contact) sports, road accidents and occupational hazards, in which head trauma causes brain damage[1]. In total, 69 million people worldwide suffer from TBI annually[2]. While TBI-induced brain damage is the leading cause of death below 45 years of age[3], nearly half of milder TBI patients experience some form of long-term cognitive impairment[4]

and are more likely to develop depression and neurodegenerative diseases[5–7]. Despite these devastating consequences for patients, there is currently a lack of specific therapeutic targets, or effective drug treatments, for TBI[8,9].

Although frequently spared from the acute primary injury, the hippocampus generally becomes affected during a secondary injury phase, which spreads throughout the brain after the initial TBI[3].

[1]Brain Plasticity Department, Swammerdam Institute for Life Sciences, Faculty of Science, University of Amsterdam, Amsterdam, The Netherlands. [2]VIB Center for Brain and Disease Research, Leuven, Belgium. [3]KU Leuven—Department of Neurosciences, Leuven, Belgium. [4]Achucarro Basque Center for Neuroscience, Sede Bldg, Campus, UPV/EHU, Barrio Sarriena S/N, Leioa, Spain. [5]Armenian Bioinformatics Institute, Yerevan, Armenia. [6]Department of Mathematics and Mechanics, Yerevan State University, Yerevan, Armenia. [7]Resolve Biosciences GmbH, Monheim am Rhein, Germany. [8]OMAPiX GmbH, Langenfeld (Rheinland), Langenfeld, Germany. [9]The Bioinformatics CRO, Niceville, FL, USA. [10]Department of Neuroscience, University of the Basque Country (UPV/EHU), Campus, UPV/EHU, Barrio Sarriena S/N, Leioa, Spain. [11]IKERBASQUE, The Basque Foundation for Science, Plaza Euskadi 5, Bilbao, Spain. [12]Instituto de Investigação e Inovação em Saúde (i3S), University of Porto, Porto, Portugal. [13]These authors contributed equally: P. Bielefeld, A. Martirosyan, S. Martín-Suárez, A. Apresyan. [14]These authors jointly supervised this work: M.G. Holt, C.P. Fitzsimons. ✉e-mail: mholt@i3S.up.pt; c.p.fitzsimons@uva.nl

Importantly, increasing evidence indicates that cognitive dysfunction after TBI is associated with changes in hippocampal function[3], that occur during this secondary phase. The hippocampus is critical for cognition, and also one of the few areas in the adult brain that harbors native neural stem cells (NSCs), that have been implicated in cognitive and emotional control[10]. Upon activation, NSCs generate proliferative progenitor cells and neuroblasts, which give rise to immature dentate granule neurons[11]. In addition, NSCs generate new astrocytes (astrogliogenesis) under physiological conditions[12–15]. These newly generated neurons and astrocytes persist in a specialized anatomical location in the subgranular zone of the Dentate Gyrus (DG), termed the adult hippocampus neurogenesis (AHN) niche[16,17]. Although the functional contribution of some of the NSC-derived neuronal cell types within the AHN niche has been studied[18,19], the properties of the adult NSC-derived astrocytes remain less well characterized[14].

Here, we aimed to assess in detail how TBI affects NSC fate in the adult hippocampus, leading to changes in neurogenesis and astrogliogenesis and, consequently, in the relative cellular composition of the AHN niche. We applied a controlled cortical impact (CCI) model of TBI[20]. The CCI model is commonly used to model moderate to severe TBI, due to its intrinsic characteristics. Frequently, an open-head CCI is used to induce reproducible local primary cortical damage and graded injury. In particular, the need for a craniotomy makes this model unsuitable for mild TBI. In the CCI model, injury severity is defined using a priori technical injury parameters, such as impact depth and speed, and a posteriori histological and behavioral outcomes, such as brain tissue loss at the site of primary injury and learning/memory impairments. Specifically, a moderate TBI induced using CCI in mice is characterized by considerable cortical tissue loss with little to no hippocampal tissue loss, and the presence of spatial learning/memory impairments[20,21]. In the moderate TBI induced by CCI, gliosis in the primary perilesional cortical region increases and then rapidly decreases to pre-impact levels shortly after impact, while hippocampal gliosis peaks later, after the first-week post impact in the in the absence of initial hippocampal tissue damage[22,23]. Interestingly hippocampal NSC proliferation and neurogenesis levels are sensitive to injury severity in the CCI model[24].

Nestin-GFP mice in which GFP is expressed in individual cells of the AHN niche under the control of the neuroepithelial stem cell protein (Nestin) promoter[25,26] has been used in single-cell RNA sequencing (scRNA-seq) studies of the AHN niche to sort NSCs, neural progenitor and other cell populations using flow cytometry, as well as to exclude that populations of NSCs are dominated by astrocytes, due to the otherwise close similarity between the two cell types[27–29]. We applied CCI to Nestin-GFP mice and analyzed the effect of TBI using Nestin-GFP cells purified by flow cytometry followed by (scRNA-seq) and spatial transcriptomics analysis of the dentate gyrus.

Here, we show that TBI disturbs the balance between NSC-driven neuro- and astrogliogenesis, effectively reducing the numbers of NSC-derived astrocytes, while increasing the number of NSC-derived neuronal cells. In addition, we molecularly characterize several cell populations derived from hippocampal NSCs. Finally, we trace back these cell populations in situ, uncovering significant changes in the anatomical location of NSC-derived cell populations in the DG after TBI. As such, our work provides a basis for future investigations of specific cell populations that could serve as targets to counteract the changes induced by TBI in the hippocampus[30], and may help us to better understand the role of NSCs in hippocampal neurodegeneration.

## Results

### TBI-induced cellular changes in the dentate gyrus and hippocampus-dependent cognition

First, we characterized cellular changes in the hippocampi of mice subjected to moderate TBI and how this correlates to changes in hippocampus-dependent cognition. For this, we compared sham craniotomized mice (Control) and mice subjected to unilateral controlled cortical impact (TBI). Fifteen days post-surgery, TBI was found to have induced unilateral hippocampal astrogliosis, as the percentage of tissue is positive for GFAP signal, a parameter that estimates astrocyte density and hypertrophy[23,31] (Fig. 1a–d). The lack of GFAP signal in the boxed area in Fig. 1a indicates the cortical tissue loss at the site of primary injury, compatible with direct cortical damage in the CCI model. We then injected mice with BrdU at the time of CCI or control craniotomy and sacrificed them 15 days after injury. We observed an increase in cell proliferation in the DG, as assessed by the number of BrdU+ cells present per mm$^3$ (Fig. 1e, f). This increase in cell proliferation was reflected in an increase in the number of local (mature) proliferating astrocytes (Fig. 1g), measured as S100B + /BrdU+ cells present per mm$^3$ (Fig. 1h) or the percentage of S100B+ cells that incorporated BrdU (Fig. 1i). Clusters of S100B + /BrdU+ cells were observed in the hilus (Fig. 1g, white arrowheads).

In addition, we observed an increased number of proliferative NSCs (Nestin-GFP + /GFAP + /Mki67+ cells) in the TBI group (Fig. 1j, k), and increased neurogenesis, as assessed by the total numbers of DCX+ cells in the DG (Fig. 1l–n). DCX+ cells where subdivided into 6 categories, according to the presence and shape of their apical dendrites: A, no processes; B, stubby processes; C, short horizontal processes; D, short vertical processes oriented to the molecular layer; E, one long vertical dendrite; F, long branched vertical dendritic tree, as described in[32]. We found a significant increase in category C and D cells (Fig. 1o), which have been classified as early neuroblasts that express the proliferation marker *Mki67*[32]. The increase in DCX+ cells in the DG of TBI mice was reflected by an increase in neuronal cell proliferation (Fig. 1p), measured as the total number of DCX$^+$/BrdU$^+$ cells per mm$^3$ (Fig. 1q) or the percentage of DCX$^+$ cells that incorporated BrdU (Fig. 1r). To address the longer-term effect of TBI on newborn neurons we traced the integration of newborn neurons in the DG using a retroviral vector expressing GFP (RV-GFP) that only labels dividing cells, which can then be visualized in hippocampal slices[33]. The RV-GFP was injected immediately before the CCI-induced TBI or control craniotomy, and mice were sacrificed 15 days after injury. We observed a significant increase in the percentage of newborn neurons (GFP$^+$ cells co-expressing DCX as a fraction of total GFP$^+$ cells) in the TBI group compared to Control (Fig. 1s, t). In addition to their numerical increase, we observed morphological changes in newborn neurons in the TBI group. Specifically, we observed that a higher percentage of the newborn neurons in the TBI group presented dendritic arbors with two primary dendrites emerging directly from the soma, or a main dendrite bifurcating within the first 10 μm from the soma (Fig. 1u). Newborn neurons in the TBI group also had a significantly lower number of dendritic spines per distance unit (Fig. 1v). The cellular changes observed in the DG corelated with impaired performance in the Morris water maze, a commonly used test for hippocampus-dependent spatial learning and memory. Mice subjected to TBI showed deficits in learning the position of a hidden platform in the test pool (Fig. 1w, x). Together, these results show that changes in the cellular architecture of the hippocampus correspond with impairments in cognition 15 days after TBI.

### A single-cell census of the AHN niche reveals that cell identity is maintained after TBI

To understand the cellular changes induced by TBI in NSCs and their progeny in the DG in a comprehensive but unbiased manner, we used scRNA-seq. Nestin-GFP mice were divided randomly into Control and TBI groups. Fifteen days post-surgery, DGs from the respective experimental groups were microdissected and pooled for tissue dissociation, followed by isolation of Nestin-GFP+ cells using fluorescence-activated cell sorting (FACS) (Supplementary Fig. 1). Cells were then subjected to scRNA-seq according to the experimental

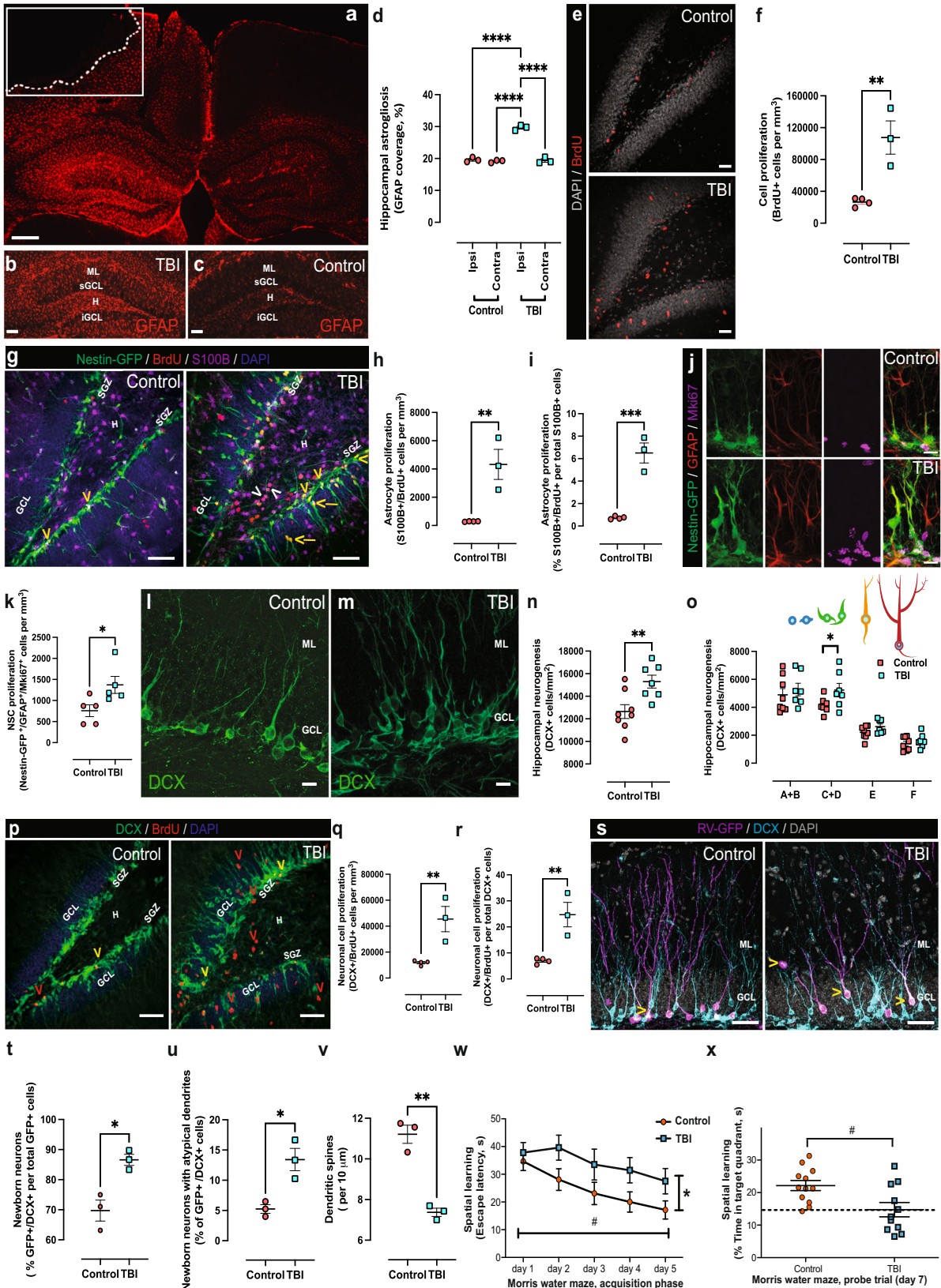

group using the 10 × 3' whole transcriptome workflow with Illumina sequencing. An unbiased integration and clustering approach was then applied to the dataset (which contained a total of 7791 high-quality cells across both experimental groups), using the Seurat algorithm[34]. We were able to identify ten cell clusters representing the expected major cell types of the DG (Fig. 2a and Supplementary Data 1). Based on

the expression levels of known marker genes, we defined abundant cell clusters containing NSCs (*Neurog2*[+], *Hmgn2*[+], *Sox4*[+], *Sox11*[+], *Mki67*[+]), Radial Glia-like (RG-like) cells (*Ascl1*[+], *Ccnd2*[+], *Vim*[+], *Hes5*[+], *Mki67*), astrocyte-like cells (*Slc1a3*[+], *Aqp4*[+], *S100b*[+], *Aldoc*[+]), neuron-like cells (*Neurod1*[+], *Snap25*[+], *Dcx*[+]), and oligodendrocytes (Oligo) (*Mog*[+], *Mag*[+], *Mbp*[+]). In addition, other less abundant cell populations were

**Fig. 1 | Immunohistochemical and behavioral characterization of hippocampal changes following TBI.** Micrograph representative of three independent experiments. Red: GFAP immunohistochemistry showing ipsilateral astrogliosis (**a**), white box: the location of the controlled cortical impact, dotted line: approximate boundary of the cortical lesion, scale bar, 250 μm. Micrographs representative of three independent experiments showing increased astrogliosis in the hippocampus of Control (**b**) vs. TBI (**c**) mice. ML molecular layer, sGCL suprapyramidal granule cell layer, H hilus, iGCL infrapyramidal granule cell layer. Scale bars, 100 μm. Quantification of GFAP surface coverage in the dentate gyrus, $n = 3$ mice. Data are presented as mean values +/− SEM, ***$P < 0.0001$ one-way ANOVA with Tukey's post hoc test (**d**). Micrographs representative of three independent experiments showing proliferative cells in the DG. Red: BrdU, gray: cell nuclei labeled DAPI, scale bars, 40 μm (**e**). Quantification of cell proliferation in the DG, $n = 4$ mice. Data are presented as mean values +/− SEM, **$P = 0.0061$ two-tailed, unpaired $t$ test (**f**). Micrographs representative of three independent experiments. Green: Nestin-GFP, red: BrdU (red), magenta: mature astrocyte marker S100B. Cell nuclei were labeled with the DNA marker DAPI (blue). Yellow arrowheads: Nestin-GFP⁺/BrdU⁺ cells, yellow arrows: Nestin-GFP/BrdU⁺ cells in the GCL. Deep pink: S100B⁺/BrdU⁺ astrocytes, white arrowheads S100B⁺/BrdU⁺ astrocytes in the hilus. Scale bars, 60 μm (**g**). Quantification of astrocyte proliferation in the DG expressed as the total number of S100B⁺/BrdU+ cells per mm³, $n = 4$ mice. Data are presented as mean values +/− SEM, **$P = 0,0060$ two-tailed, unpaired $t$ test (**h**). Quantification of astrocyte proliferation in the DG expressed as the % of total S100B⁺ cells that incorporated BrdU, $n = 4$ mice. Data are presented as mean values +/− SEM, ***$P = 0,0063$ two-tailed, unpaired $t$ test (**i**). Micrographs representative of three independent experiments showing proliferative NSCs in the DG (Nestin-GFP⁺/GFAP⁺/Mki67⁺ cells). Green: Nestin-GFP, red: GFAP, magenta: Mki67, merge: Nestin-GFP⁺/GFAP⁺/Mki67⁺, scale bars, 15 μm (**j**). Quantification of NSC proliferation in the DG expressed as the total number of Nestin-GFP⁺/GFAP⁺/Mki67⁺ per mm³, $n = 5$ mice. Data are presented as mean values +/− SEM, *$P = 0.0374$ two-tailed, unpaired $t$ test (**k**). Micrographs representative of three independent experiments showing cells expressing DCX in the DG of Control (**l**) or TBI (**m**) mice, scale bars, 10 μm. Quantification of hippocampal neurogenesis expressed as total numbers of DCX+ cells in the dentate gyrus, Control $n = 5$ mice, TBI $n = 4$ mice. Data are presented as mean values +/− SEM, *$P < 0.0078$ two-tailed unpaired $t$ test (**n**). Quantification of six DCX+ cell phenotypes according to the presence, shape and orientation of apical dendrites [27] within the DG. Control $n = 5$ mice, TBI $n = 4$ mice. Data are presented as mean values +/− SEM, *$P = 0.0431$ one-way ANOVA with Tukey's post hoc test (**o**). Micrographs representative of three independent experiments showing proliferative cells in the DG, Green: DCX antibodies, red: BrdU, blue: cell nuclei labeled DAPI. Yellow somas indicated by yellow arrowheads: DCX⁺/BrdU⁺ cells, red somas indicated by red arrowheads: (DCX⁻/BrdU⁺) cells, scale bars, 80 μm (**p**). Quantification of neuronal cell proliferation in the DG expressed as the total number of DCX⁺/BrdU⁺ cell per mm³, $n = 4$ mice. Data are presented as mean values +/− SEM, **$P = 0.0095$ two-tailed, unpaired $t$ test (**q**). Quantification of neuronal cell proliferation in the DG expressed as the % of total DCX⁺ cells that incorporated BrdU, $n = 4$ mice. Data are presented as mean values +/− SEM, **$P = 0.0068$ two-tailed, unpaired $t$ test (**r**). Micrographs representative of three independent experiments. Magenta: RV-GFP, cyan: DCX. Yellow arrows: RV-GFP⁺/ DCX⁺ cells in the external half of the GCL or the ML, scale bars, 20 μm (**s**). Quantification of newborn neuron numbers expressed as the % of DCX+ cells that were positive for RV-GFP, $n = 4$ mice. Data are presented as mean values +/− SEM, *$P = 0.0129$ two-tailed, unpaired $t$ test (**t**). Quantification of the number of newborn neurons with two primary dendrites emerging directly from the soma, or a main dendrite bifurcating within the first 10 μm from the soma (atypical dendrites) expressed as the % of total newborn neurons, $n = 4$ mice. Data are presented as mean values +/− SEM, *$P = 0.0146$ two-tailed, unpaired $t$ test (**u**). Quantification of dendritic spines density in secondary and tertiary dendrites of newborn cells expressed as the number of spines per 10 μm long dendritic segment, $n = 4$ mice. Data are presented as mean values +/− SEM, **$P = 0.0014$ two-tailed, unpaired $t$ test (**v**). Escape latency in the Morris Water Maze test Control $n = 12$ mice, TBI $n = 11$ mice. Data are presented as mean values +/− SD, #$F(1, 109) = 67,0012$, $P = 5,56 \cdot 10^{-13}$ time, *$F(4, 109) = {}^{1}9,6203$, $P = 3,47 \cdot 10^{-12}$ treatment, two-way repeated measures ANOVA with Bonferroni post hoc test. Exact $P$ values were calculated using $P = F.DIST.RT(F, DFn, DFd)$ (**w**). Percentage of time spent in the target quadrant during the Morris water maze probe trial, Control $n = 12$ mice, TBI $n = 11$ mice. Data are presented as mean values +/− SEM, #$P = 0.011$ two-tailed, unpaired $t$ test (**x**). Source data are provided as a Source Data file.

identified, such as oligodendrocyte precursor cells (OPC) (*Pdgfra⁺, Gpr17⁺, Mag⁺*), pericytes/mural cells (Mural) (*Des⁺, Col1a2⁺*), endothelial cells (Endo) (*Pecam1⁺, Flt1⁺*) and microglia (Mgl) (*Cd68⁺, Cx3CR1⁺*). Crucially, UMAP representation indicated that TBI does not appear to induce major changes in the identifiable cell types present in the DG (Fig. 2b), although increases in oligodendrocyte, endothelial, microglia, and mural cell populations were detected, alongside decreases in NSC and astrocyte populations. In contrast, despite the neuronal population appearing to increase slightly consistent with increased neurogenesis following TBI, this change was not considered statistically significant with the threshold for statistical significance of $P < 0.05$ used for other populations (Fig. 2c and Supplementary Data 1), although this may reflect the issue of sampling across the total neuronal population.

Based on the known developmental origin of neurons and astrocytes from radial glia-like NSCs in the hippocampus[14,35], and our own observation of alterations in both astrocyte and neuron populations following TBI (Fig. 1), we decided to examine the effects of TBI on neurogenesis and astrogliogenesis in greater detail. For this, the NSC, RG-like, astrocyte-like and neuron-like populations were extracted, and reclustering was performed. This approach revealed several populations (Fig. 2d and Supplementary Data 2), including three clusters with gene expression profiles characteristic of NSCs. These three NSC clusters expressed known NSC markers (*Ascl1, Vim, Hes5, Id4*) (Supplementary Fig. 2), and could be further differentiated based on differential marker expression: NSC-stage 1 (expressing *Ranbp1, Ezh2, Nme1*), NSC-stage 2 (expressing *Eif4g2, Hspa5*) and RG-like (expressing *Neurog2, Zeb 1*) (Fig. 2d, Supplementary Fig. 3, and Supplementary Data 3). Importantly, NSC-stage 1 and 2 cells robustly expressed the proliferation marker *Mki67*, but only a few RG-like cells did (Supplementary Fig. 3 and Supplementary Data 3). To investigate

the neurogenic and astrogliogenic pathways in more detail, we performed a pseudotime lineage analysis on the data using the Monocle algorithm[36], with the identified NSC populations as the developmental root. Using this methodology, neuronal populations could be divided in UMAP space into a neuronal lineage which included NSC-stage 1 and 2, RG-like cells and other clusters that expressed *Dcx* and *NeuroD1, Mdk, Prox1* (N-stage 1–2 cells); *Plxna4, NeuroD2, Prox1* (N-stage 3–4 cells); *Tubb2b, Fez1, Prox1* (N-stage 5–6, cells) (Supplementary Fig. 3 and Supplementary Data 3). N-stage 7 cells, expressing *Syt1 and Reln* clustered independently from N-stage 1–6 cells in UMAP space, indicating they likely belong to a different lineage not derived from NSCs (Fig. 2d, Supplementary Fig. 3, and Supplementary data 3). Pseudotime analysis also revealed an astrocytic lineage including NSC-stage 1 and 2, RG-like and four astrocytic cell clusters which we termed A-stage 1–4 cells. A-stage 1 cells expressed NSC markers (*Ascl1, Vim, Hes5*) and genes commonly expressed in astrocytic cells (*Gfap, Ntrk2, Id2*), suggesting A-stage 1 cells are immature astrocytic cells originating from NSCs. A-stage 2 cells did not express the NSC markers *Ascl1, Vim*, and *Hes5*, but expressed the astrocytic markers *Gfap, Ntrk2 and Id2*, presumably representing cells differentiating along the astrocytic lineage. A-stage 3–4 cells expressed *Gfap* but differed from A-stage 2 cells by expressing a different set of markers, including *Smo, Fgfr3, Dmd*, and *Hes5*. A-stage 5–7 cells clustered separately in UMAP space, as they neither expressed NSC markers (*Ascl1, Hes5*) nor the astrocytic markers *Smo, Fgfr3*, and *Dmd*. They did, however, express *Gfap, Ntrk2*, and *Id2*, indicating that they likely represent a different astrocyte lineage that is not derived from NSCs (Fig. 2d, Supplementary Fig. 3, and Supplementary Data 3). Interestingly, TBI did not induce significant changes in the overall developmental trajectories of NSC-derived neurons and astrocytes (Fig. 2e). However, several NSC-derived neuronal cell clusters (N-stage 1, 2, 4, and 5) were enriched in the cell suspensions from

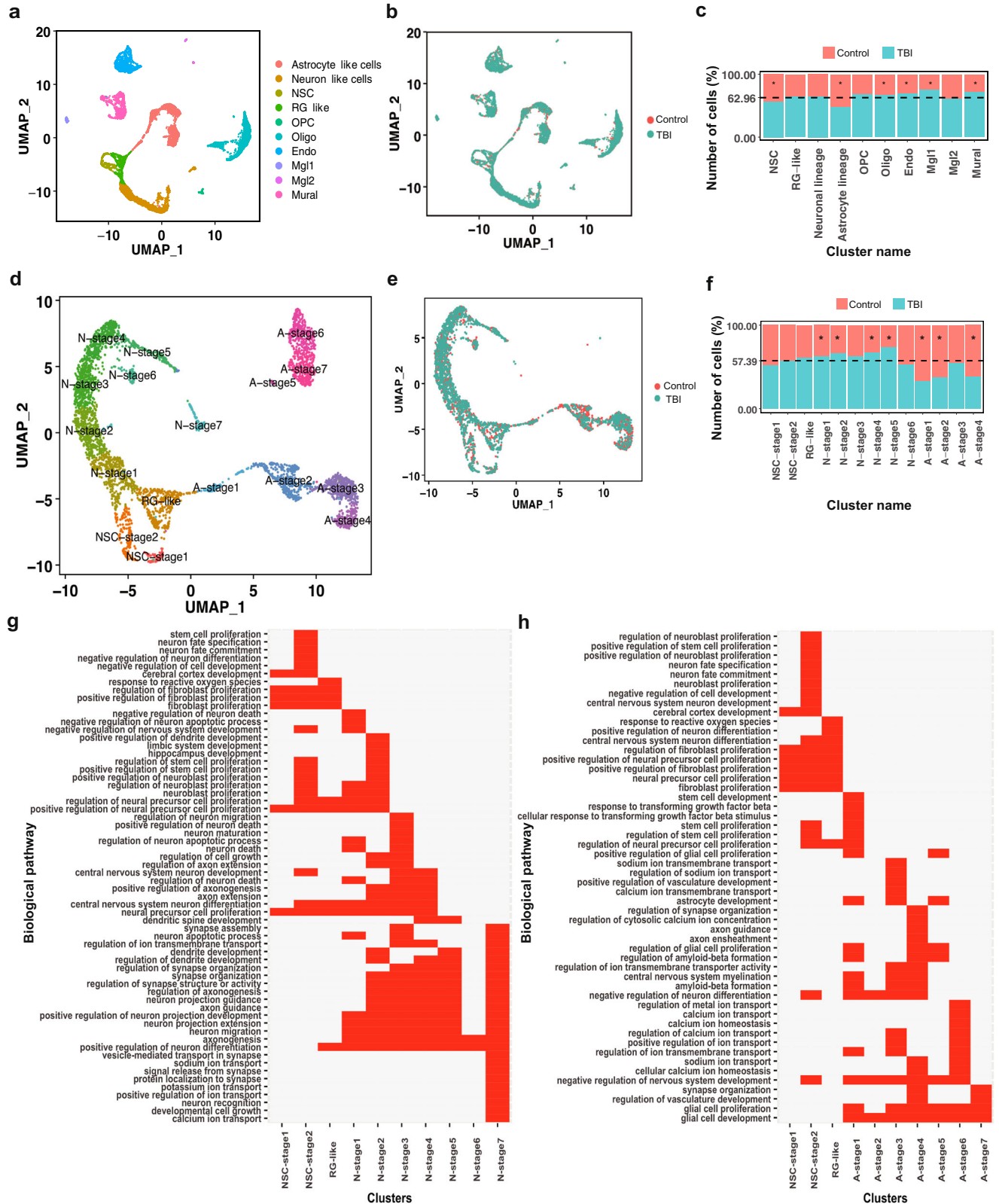

the TBI group, with N-stage 5 cells showing the largest TBI-induced increase (Fig. 2f and Supplementary Data 2). In addition, several NSC-derived astrocytic cell clusters were depleted in these cell suspensions, with A-stage 1, 2, and 4 cells showing a significant TBI-induced reduction, while A-stage 3 cells were unaffected (Fig. 2f and Supplementary Data 2). In the absence of major effects of TBI on lineage and cell-type identity, the change in the relative proportions of the various neuronal

and astrocyte subgroups suggests that TBI affects principally cell fate determination. Gene ontology (GO) biological pathway (BP) analysis performed for the neuro- and astrogliogenic lineages confirmed two trajectories, with distinct changes in associated biological functions over pseudotime, indicating a gradual change towards neuronal (Fig. 2g and Supplementary Data 4 and 5) or glial (Fig. 2h and Supplementary Data 4 and 5) functions, respectively. Interestingly,

**Fig. 2 | Molecular characterization of the NSC-derived neuro- and astro-gliogenic lineages within the mouse dentate gyrus and the effects of TBI.** UMAP-based visualization of the major higher-order cell types identified by Seurat in the single-cell dataset. Each dot represents a single cell. Cells with similar molecular profiles group together. Cell types were assigned according to the expression of specific marker genes and are labeled in different colors (**a**). UMAP-based visualization of single-cell data according to experimental origin (Control or TBI, 5–6 animals per condition) (**b**). Bar plots showing the relative number of cells per cluster originating from Control or TBI samples against their predicted abundance (62.96% of all cells originating from TBI samples: dashed line), NSC, *$P = 4.69 \times 10^{-2}$; Astrocyte linage, *$P = 5.99 \times 10^{-35}$; Oligo, *$P = 2.92 \times 10^{-3}$; Endo, *$P = 5.14 \times 10^{-5}$; Mgl1, *$P = 2.13 \times 10^{-2}$; Mural, *$P = 1.82 \times 10^{-8}$ vs. expected abundance, two-sided binomial test. A summary of cell numbers and statistical analyses performed is given in Supplementary Data 1 (**c**). UMAP-based visualization of NSC-derived neuronal and astrocytic lineages obtained by extraction of neuronal and astrocytic cells followed by reclustering. Each dot represents an individual cell and discrete cell states are identified in specific colors (**d**). UMAP-based visualization of reclustered data according to experimental origin (Control or TBI) (**e**). Bar plots showing the relative number of cells per identified cell cluster in Control or TBI samples, against their expected abundance (57.39% of all cells originating from TBI samples: dashed line), N-stage 1, *$P = 4.66 \times 10^{-2}$; N-stage 2, *$P = 4.07 \times 10^{-4}$; N-stage 4, *$P = 2.97 \times 10^{-4}$; N-stage 5, *$P = 2 \times 10^{-5}$; A-stage 1, *$P = 2.23 \times 10^{-7}$; A-stage 2, *$P = 3.39 \times 10^{-15}$; A-stage 4, *$P = 2.12 \times 10^{-6}$ vs. expected abundance, two-sided bino-mial test. A summary of cell numbers per population and statistical analyses performed is given in Supplementary Data 2 (**f**). GO biological pathway matrix for the cell clusters included in the neuronal (**g**) and astrocytic (**h**) lineages. Source data are provided as a Supplementary Data file.

GO analysis indicated NSC-stage 2 is the cell population with the highest proliferative potential, suggesting these cells represent activated NSCs, in agreement with their robust *Mki67* expression (Supplementary Figs. 2 and 3 and Supplementary Data 3). A previous study[37], identified two astrocyte populations in the hippocampus, AST4 and AST5, with potential NSC or intermediate progenitor characteristics. Using an integrated analysis of hierarchical clustering, pseudotime profiles, and GO functional annotations for cell populations based on their gene expression profiles (Supplementary Fig. 4), we compared the astrocytic cell types identified in our study with AST4 and AST5. AST4 clustered closely with RG-like, NSC-stage 1 and NSC-stage 2 cells (Supplementary Fig. 4). AST5 clustered closely to A-stage 6 and A-stage 7 cells (Supplementary Fig. 4), which did not fit within the astrocytic pseudotime trajectory derived from NSCs (Fig. 2d, e), indicating a different origin. Similarly, AST4 displayed a pseudotime trajectory comparable to our RGL, NSC-stage-1 and NSC-stage-2 cells, while AST5 pseudotime trajectory differed from those (Supplementary Fig. 4). Finally, DAVID functional GO analysis of the genes expressed in AST4 revealed several biological pathways associated with translation, resembling BPs such as "cytoplasmatic translation", "ribosome assembly" and "ribosome biogenesis" overrepresented in RG-like, NSC-stage1 and NSC-stage2 (Supplementary Fig. 5). In addition, GO pathway analysis indicated that the BP "glial cell proliferation" was over-represented in several NSC-derived astrocytic cell clusters that resemble AST5, supporting the hypothesis they represent an intermediate transition state between progenitors and mature astrocytes[37].

## RNA velocity predicts NSC fate changes after TBI

RNA velocity is a computational tool for predicting future cell state from scRNA-seq data, by analyzing the ratio of spliced versus unspliced RNA transcripts[38,39]. RNA velocity analysis performed on our scRNA-seq dataset confirmed neuronal and astrocytic lineages originating from NSCs in the AHN niche, in Control and TBI conditions (Fig. 3a, b). In these RNA velocity plots, the direction of the RNA velocity vector indicates the likely future state of a given cell[40], with long vectors corresponding to large changes in gene expression indicative of cells actively undergoing state transitions and short vectors representing cells having low-transition probabilities and therefore largely maintaining homeostasis[38,41]. Based on the calculated transition probabilities, it appears both NSC-stage 1 and NSC-stage 2 cells transition to RG-like cells in Control conditions (orange bars Fig. 3c, d, respectively), which in turn self-renew and feed both the neuronal (N-stage 1) and astrocytic (A-stage 1) lineages (orange bars Fig. 3e and Supplementary Data 6). Crucially, RG-like cells were the only cell type predicted to generate astrocytic progeny in both experimental groups (Fig. 3a, b and Supplementary Data 6). Consistent with the numerical changes observed in specific cell clusters after TBI (Fig. 2f), RNA velocity indicated an increased probability of NSCs entering the neuronal lineage following injury, which is also in line with the increase in immature neurons detected using RV-GFP tracing and immunostaining (Fig. 1).

Our RNA velocity calculations indicated and increased probability that NSC-stage-2 and RG-like cells transition to N-stage 1 (Fig. 3d, e, respectively and Supplementary Data 6), as well as NSC-stage 1 cells transitioning to NSC-stage 2 (Fig. 3c and Supplementary Data 6). In contrast, the probability of RG-like cells progressing to A-stage 1 was reduced in the TBI group (Fig. 3e and Supplementary Data 6), indicating that increased neurogenesis likely occurs at the expense of astrogliogenesis. Interestingly, in Control conditions, a proportion of NSC-stage-2 cells showed a likelihood of transitioning to N-stage-1 cells, an effect promoted in TBI conditions (Fig. 3d and Supplementary Data 6), further tipping the balance of cell production toward neurogenesis. Consistent with an imbalance in cell numbers, TBI had a strong effect on the apparent self-renewal behaviors, observed as retention of state identity in our transition probability analysis, of both NSC-stage-1 and RG-like cells. TBI disfavored the apparent self-renewal of NSC-stage 1 and increased that of RG-like cells (Fig. 3c, e and Supplementary data 6). Moving downstream of the initial neurogenic and astro-gliogenic events leading to N-stage 1 and A-stage 1 cells, respectively, RNA velocity showed that TBI promoted the transition of N-stage 1 cells to N-stage 2 cells and the transition of A-stage 1 cells to A-stage 2 cells (Supplementary Fig. 6). In addition, we observed a dynamic "back flow" within cells of the astrocytic lineage. A-stage 3 and A-stage 4 cells appear able to transition to A-stage 2 cells with high probability in Control conditions, and these transitions were disfavored in TBI conditions (Supplementary Fig. 6). A-stage 4 cells could also transition to A-stage 3 cells in Control, and this transition was promoted in TBI conditions (Supplementary Fig. 6). In summary, our RNA velocity analysis is consistent with the major effect of TBI being on cell fate decisions/transitions rather than cell identity per se.

## Differential gene expression analysis reveals cell population-specific responses to TBI

Differential gene expression analysis in combination with GO pathway analysis is a commonly used approach to infer changes in cellular processes and functions from scRNA-seq datasets[42]. Here, we used this approach to interpret the changes induced by TBI in specific cell populations derived from hippocampal NSCs. First, we compared gene expression patterns per cell population between the Control and TBI groups using the two-sided MAST test. We found several differentially expressed genes (DEGs) in specific cell populations. Overall, TBI resulted in more upregulated genes (38 in total across 9 cell populations) than downregulated genes (16 in total across 8 cell populations) (Fig. 4a, b and Supplementary Data 7). A-stage 3 was the cell population with the largest number of DEGs, suggesting A-stage 3 is the population most affected by TBI in terms of changes in cell function. Interestingly, we found that TBI upregulated a group of genes generally associated with neuronal development and function that are commonly repressed during astrocytic differentiation (Sox4, Stmn1, Eid1, Pcp4) (Fig. 4a). This suggests that TBI may affect the differentiation status of A-stage 3 cells. In agreement with the former interpretation,

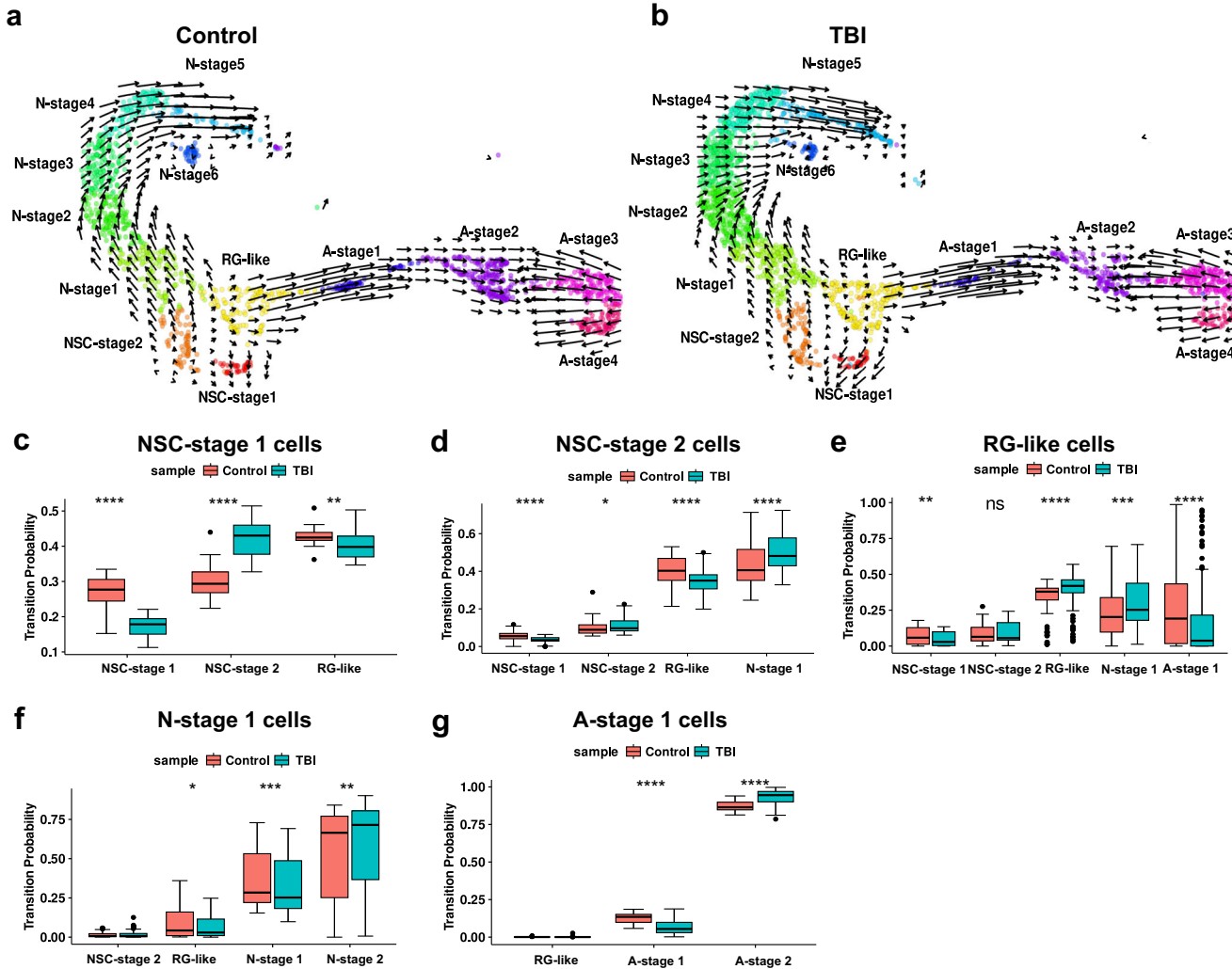

**Fig. 3 | RNA velocity analysis indicates that TBI induces a subtle but functionally significant shift in hippocampal neural stem cell fate.** RNA velocity analysis along both the neuronal and astrocytic lineage in the Control (**a**) and TBI (**b**) groups, vectors (arrows) indicate the predicted direction and speed of single-cell transitions in transcriptome space, colors indicate previously characterized cell clusters indicated in the figures. Box blots showing transition probabilities calculated for NSC-stage 1 cells to NSC-stage 1 cells (****$P$ = 4.4 × 10$^{-15}$), to NSC-stage 2 (****$P$ = 3.7 × 10$^{-15}$) and to RG-like cells (**$P$ = 1.5 × 10$^{-3}$) (**c**); NSC-stage 2 cells to NSC-stage 1 (****$P$ = 6.5 × 10$^{-8}$), to NSC-stage 2 (*$P$ = 1.7 × 10$^{-2}$), to RG-like (****$P$ = 1.4 × 10$^{-5}$) and N-stage 1 cells (****$P$ = 7.9 × 10$^{-5}$) (**d**); RG-like cells to NCS-stage 1 (**$P$ = 1.2 × 10$^{-3}$), to RG-like (****$P$ = 6.6 × 10$^{-12}$), to N-stage 1 (****$P$ = 9.3 × 10$^{-4}$) and to A-stage 1 cells (****$P$ = 3.6 × 10$^{-6}$) (**e**); N-stage 1 cells to RG-like (*$P$ = 4.7 × 10$^{-2}$), to

N-stage 1 (***$P$ = 7.1 × 10$^{-4}$), and to N-stage 2 cells (**$P$ = 1.7 × 10$^{-3}$) (**f**); and A-stage 1 cells to A-stage 1 (****$P$ = 5.4 × 10$^{-9}$) and to A-stage 2 cells (****$P$ = 3.9 × 10$^{-9}$) (**g**). Orange bars: Control group, cyan bars: TBI group, using Control = 5 and TBI = 6 animals per condition. In all panels, independent non-parametric two-sided Wilcoxon test, Control vs TBI. Box plots show the median, first quartile (25%), third quartile (75%), and interquartile range. Whiskers represent data minima and maxima; dots are data points located outside the whiskers, $n$ = number of cells per cell subpopulation Control/TBI are available as Supplementary Data 2 and calculated mean transition probabilities across cell clusters are available in Supplementary Data 14, including exact $P$ values for all comparisons. Source data are provided as Supplementary Data 15.

we found BP related to "positive regulation of neuron differentiation", "positive regulation of neuron projection development" and "central nervous system projection neuron axogenesis" significantly over-represented in A-stage 3 cells after TBI (Supplementary Data 8 and 9). In addition, we found that TBI resulted in downregulation of two genes involved in the negative regulation of cell cycle progression (Arhgdia, Nacc2) in A-stage 3 cells, suggesting that proliferation of A-stage-3 cells may be favored after injury. TBI did not, however, induce large changes in gene expression in NSC populations. In RG-like cells, two genes were found upregulated: mediator complex subunit 29 (*Med29*) and protein phosphatase 1 regulatory inhibitor subunit 14B (*Ppp1r14b*). *Ppp1r14b*, is a gene linked to cell proliferation, growth and apoptosis[43–45], and *Med29* is amplified and overexpressed in hyperproliferative cells[45–47], suggesting that upregulation of these two genes may be involved in

proliferative changes and fate changes in RG-like cells (consistent with RNA velocity predictions). Based on this finding in RG-like cells, a detailed examination of DEGs across cell populations revealed that *Ppp1r14b* was consistently upregulated following TBI in 7 cell populations (Fig. 4a and Supplementary Data 7), specifically RG-like (Fig. 4c), NSC-stage 2, (Fig. 4d), astrocytic A-stage 3 (Fig. 4e), and neuronal N-stage 1–4, (Fig. 4f–i) populations, suggesting that *Ppp1r14b* may play an important role in the response of these cell populations to TBI.

## TBI affects the anatomical location of specific NSC-derived populations in the DG

Our scRNA-seq results indicate the presence of previously uncharacterized cell populations in the DG. To validate their existence, we performed multiplexed fluorescence in situ hybridization (FISH) using

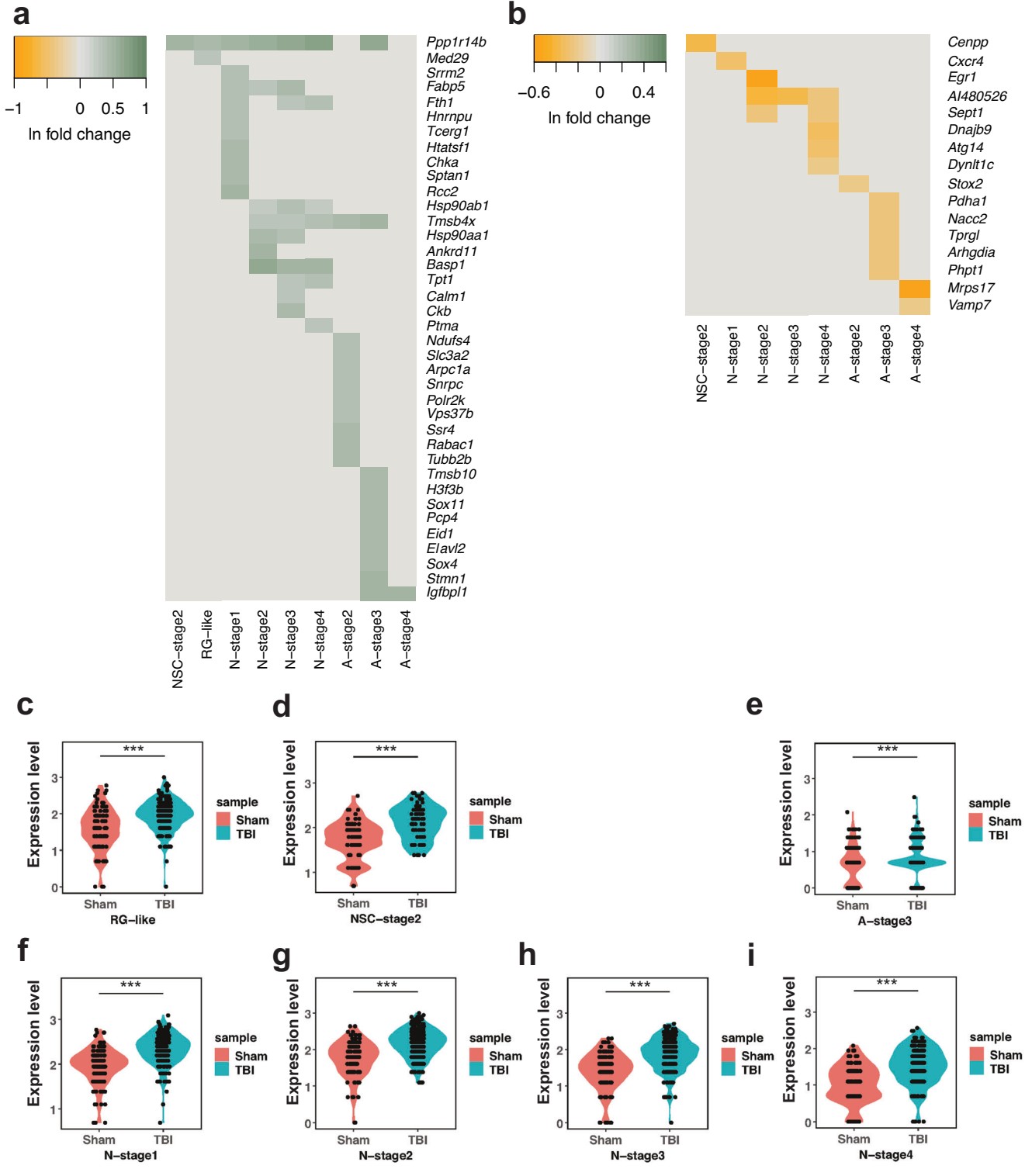

**Fig. 4 | TBI induces cell population-specific changes in gene expression in NSCs and NSC-derived cells in the DG.** Heatmaps showing expression of individually upregulated (**a**) and downregulated (**b**) genes across NSCs and NSC-derived cell populations. Color bars indicate the relative intensity of expression, indicated as ln fold change in TBI vs. Control groups; SCT normalized data are compared. Violin plots showing changes in *Ppp1r14b* expression induced by TBI in RG-like cells,

***$P = 5 \times 10^{-4}$ (**c**), NSC-stage 2 cells, ***$P = 8.28 \times 10^{-7}$ (**d**) astrocytic A-stage 3 cells, ***$P = 4.93 \times 10^{-5}$ (**e**), and neurogenic N-stage 1, ***$P = 3.21 \times 10^{-15}$ (**f**), N-stage 2, ***$P = 8.59 \times 10^{-16}$ (**g**), N-stage 3, ***$P = 1.38 \times 10^{-13}$ (**h**) and N-stage 4 cells, ***$P = 6.44 \times 10^{-12}$ (**i**), two-sided MAST test with Bonferroni post hoc test on SCT normalized data. Source data are provided at https://zenodo.org/records/10829090/files/AstrocyticAndNeuronalLineage_10x.RData?download=1.

RNAscope[48] in hippocampal slices from wild-type mice, focusing on the identification of NSC-derived astrocytic cells. For this, we used set of 7 probes targeting *Slc1a3, Hapln1, Frzb, Ascl1, Sparc, Sned1,* and *Neat1* (Supplementary Fig. 7, "Methods"). Using this strategy, we were

able to localize three of the astrocytic cell populations in the DG identified in our scRNA-seq experiments (Fig. 5a): A-stage 4 cells (*Slc1a3+, Hapln1+, Neat1+, Sned1+, Sparc−, Ascl1−, Frzb−*; Fig. 5a inset iii, Supplementary Fig. 7), A-stage 2 cells (*Slc1a3+, Hapln1−,*

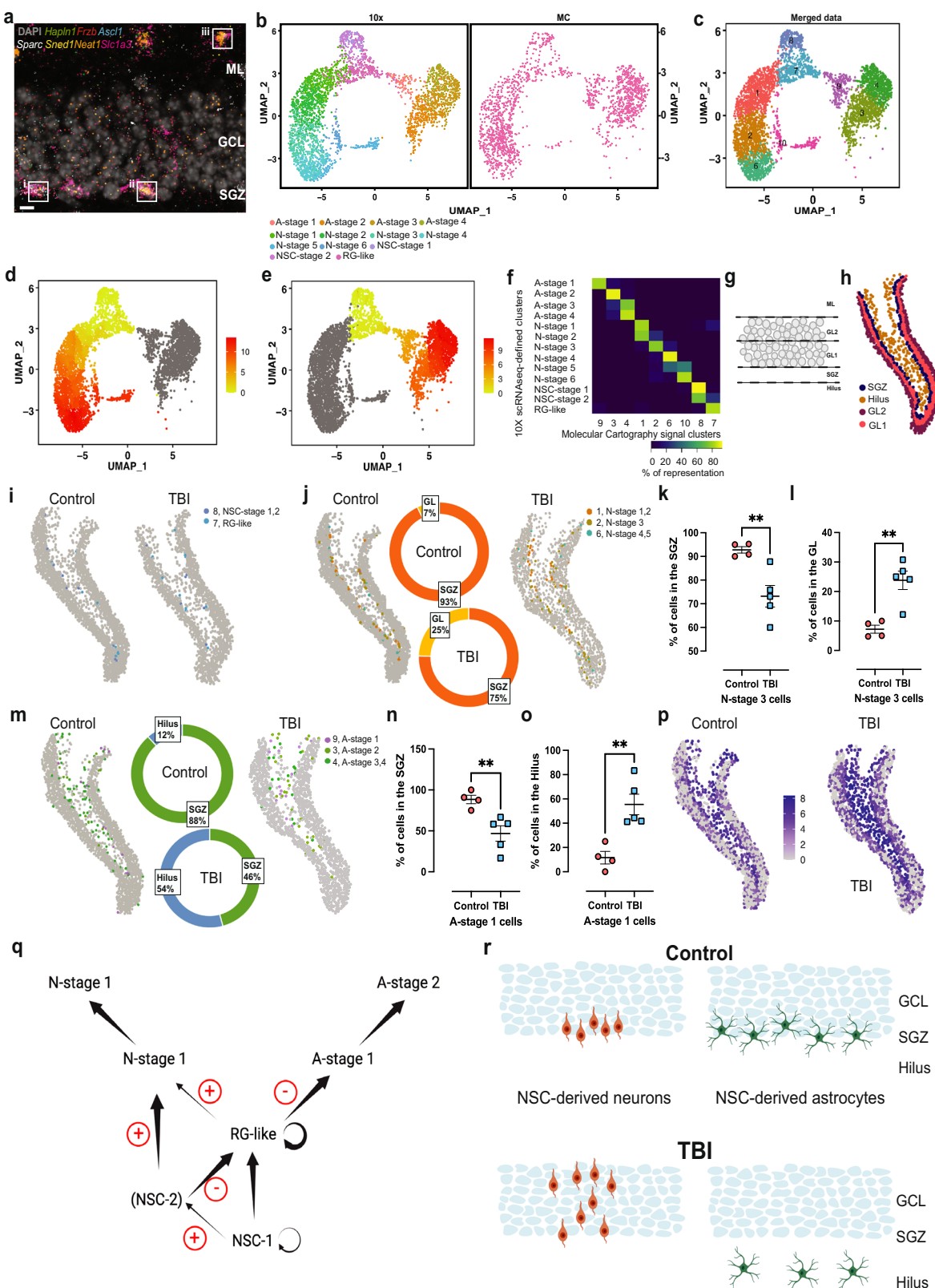

*Neat1+, Sned1+, Sparc−/+, Ascl1−, Frzb−*; Fig. 5a, inset ii and Supplementary Fig. 7) and A-stage 1 cells (*Slc1a3+, Hapln1−, Frzb+, Ascl1+, Sparc+, Neat1+, Sned1+*, Fig. 5a, inset I and Supplementary Fig. 7). These different cell populations were observed in distinct locations within the DG. A-stage 1 and 2 cells, were located in the subgranular zone (SGZ) (Fig. 5a, insets i and ii, respectively), while A-stage 4 cells were located in the molecular layer (ML), close to the external layers of

the granule cell layer (GCL) (Fig. 5a, inset iii). Indeed, NSCs and their neuronal and astrocytic progenies are found in the SGZ, and this anatomical positioning has been linked to their functional roles and integration into hippocampal circuits[14,18,49]. Overall, our results with RNAscope confirmed that early NSC-derived astrocytic cells (A-stage 1 and 2) located to the SGZ, while other populations in the NSC-derived astrocytic lineage may reside in more distant locations. Unfortunately,

**Fig. 5 | TBI induces changes in the location of specific cell populations in the dentate gyrus.** Micrograph representative of three independent experiments showing RNAscope visualization of marker genes for A-stage 4 (**a**, inset iii), A-stage 2 (**a**, inset ii) and A-stage 1 (**a**, inset i) cells (individual marker signals shown in Supplementary Fig. 7). ML molecular layer, GCL granule cell layer, SGZ subgranular zone; scale bar, 5 μm. UMAP-based representation of the individual 10X and Molecular Cartography (MC) datasets. scRNA-seq-derived (left) and MC-derived data (right) share a similar distribution within UMAP space. Colors in the 10X dataset indicate neuronal and astrocytic cell lineages (**b**). UMAP-based representation of the combined 10X-MC dataset (merged data), with colors indicating the ten distinct clusters identified (**c**). UMAP-based representation of pseudotime for neuronal (**d**) and astrocytic (**e**) lineages using the combined 10X-MC dataset. Color bars: relative pseudotime distance from the root population (yellow: minimum, red: maximum). Correlation matrix showing the relative overlap in gene expression data between cell populations identified by scRNA-seq (10X scRNA-seq-defined clusters) and MC (Molecular Cartography signal clusters) (**f**). Color bar: % of genes in a scRNA-seq cluster present in a MC cluster. DG subdivisions used to compare the location of MC clusters, Hilus, subgranular zone (SGZ), inner granule cell layer (GL1), outer granule cell layer (GL2). ML is indicated as reference (**g**). Spatial mapping for an example Control DG section (**h**). Spatial mapping of NSC populations in Control and TBI groups in the DG. Colored dots indicate the location of single-cell populations indicated in the legend (**i**). Spatial mapping of NSC-derived neuronal cells in Control and TBI groups in the DG. Colored dots indicate

the location of single-cell populations indicated in the legend. Pie charts: % localization of N-stage 3 cells to the SGZ (orange) or the GL (yellow) (**j**). Dot plots showing quantification of N-stage 3 cells in the SGZ (**k**) or the GL (**l**), $n = 4$ Control, $n = 5$ TBI. Data are presented as mean values +/− SEM, **$P = 0.0075$ (**k**), $P = 0.0030$ (**l**), two-tailed unpaired $t$ test. Spatial mapping of NSC-derived astrocytic populations in Control and TBI groups. Colored dots indicate the location of single-cell populations indicated in the legend. Pie charts: % localization of astrocytic A-stage 1 cells to the SGZ (green) or the Hilus (blue) (**m**). Dot plots showing the quantification of A-stage 1 cells in the SGZ (**n**) or the Hilus (**o**) $n = 4$ Control, $n = 5$ TBI. Data are presented as mean values +/− SEM, **$P = 0.0088$ (**n**), $P = 0.0043$ (**o**), two-tailed unpaired $t$ test. Spatial mapping of *Gfap*+ astrocytes in the DG from Control and TBI animals (**p**); dots indicate the location of single cells, with color encoding the relative intensity of *Gfap* expression. Spatial representations in (**j**, **m**) are representative examples of hippocampal slices analyzed to generate the data in (**k**, **l**, **n**, **o**). Summary of cell transitions predicted by RNA velocity in the Control and TBI groups. Black arrows indicate the most probable transitions of NSCs and NSC-derived cells along neuronal and astrocytic lineages in uninjured AHN; arrow thickness indicates transition probability. Red +: transition promoted by TBI; Red −: transition inhibited by TBI. Circular arrows: cell populations with predicted self-renewal potential (**q**). Representations of GCL/SGZ/Hilus areas in the DG indicating the location of NSC-derived neuronal or astrocytic cells determined using MC spatial transcriptomics in both Control and TBI (**r**). Source data are provided as a Source Data file.

RNAscope is restricted to the simultaneous detection of 12 transcript probes. To overcome this limitation, we aimed to characterize the anatomical location of NSCs and NSC-derived neurogenic and astro-cytogenic cells using the recently developed Molecular Cartography spatial transcriptomics platform (Resolve Biosciences GmbH, Germany)[50]. Molecular Cartography allows for simultaneous detection of up to 100 transcripts. We designed an extended panel of 93 probes to localize cell populations identified from our scRNA-seq dataset in the DG of mice from Control and TBI groups (Supplementary Fig. 7 and Supplementary Data 10). In a first validation step, the localization of probes in Control tissue was compared to the corresponding signals obtained from the Allen Brain Atlas[51], which showed matching patterns (Supplementary Fig. 7). Using the anchor-integration method in the Seurat package, we then compared the signals from the 93 Molecular Cartography targets to the scRNA-seq dataset. This analysis indicated a substantial overlap in the cell populations identified by both techniques (Fig. 5b, c), with all NSC and NSC-derived populations defined in the scRNA-seq data found back in the Molecular Cartography signal clusters (MCCs) (Fig. 5c). Supporting the high concordance between both techniques, pseudotime analysis of the merged scRNA/MCCs dataset confirmed the presence of NSC-derived neurogenic and astrocytogenic lineages (Fig. 5d, e). Although the relationship between the two datasets was not always one-to-one, as some of the populations defined by scRNA-seq were represented in more than one MCC (Fig. 5f), two pieces of evidence suggest that the two datasets report equivalent features. First, there is a high degree of overlap between the datasets. Second, an analysis of the percentual contribution of cells to each MCC per experimental condition, showed that the number of cells in MCC 1 (N-stage 1 and 2 cells, Fig. 5f), was significantly larger in the TBI group and the number of cells in MCC 3 and 4 (A-stage 2–4 cells, Fig. 5f) was significantly smaller in the TBI group (Supplementary Fig. 7 and Supplementary Data 13), demonstrating a high degree of correspondence between the scRNA-seq and Molecular Cartography datasets with respect to the effect of TBI on the relative cellular composition of the DG. Therefore, we then asked whether TBI affects the location of specific cell populations within the DG. First, we segmented the DG into four different regions: SGZ, inner GCL (GL1); outer GCL (GL2), and Hilus, as described in refs. 52,53. Guided by the DAPI signal indicating cell nuclei in the GCL, the SGZ was defined as a two-nucleus-wide band below the GCL, while the GL2 was defined as a three-nucleus-wide band below the Molecular Layer. Based on these criteria the GCL1 can be considered as the band between the SGZ and

the GCL2, and the Hilus the area below the SGZ. As the thickness of the GCL ranges from 4 to 8 neurons[54], this procedure divides the GCL in two halves of similar size (Fig. 5g, h). Then, we compared the location of NSCs and the corresponding neuronal and astrocytic lineages in the Control and TBI groups, using the MCCs defined in Fig. 5f. We were unable to detect any change induced by TBI in the location of cells in MCCs 7 and 8 (RG-like, NSC-stage 1 and NSC-stage 2 cells) (Fig. 5i). However, the location of cells in MCC 2 (N-stage 3 cells) and MCC 9 (A-stage 1 cells) was affected by TBI (Fig. 5j–o). The percentage of N-stage 3 cells in the SGZ was reduced (Fig. 5j, k), while their number in the GCL (GL1 + 2) was increased (Fig. 5j, l) in the TBI group. The percentage of A-stage 1 cells in the SGZ was reduced (Fig. 5m, n), while their number in the Hilus was increased (Fig. 5m, o) in the TBI group. These results indicate a specific effect of TBI on the location of two NSC-derived cell populations in the DG, displacing them away from their native locations in the SGZ. Interestingly, Molecular Cartography showed an increase in *Gfap*+ cells in the hilus (Fig. 5o), in agreement with the general increase in GFAP protein levels observed in the hippocampus in the TBI group (Fig. 1a–d).

## Discussion

Under physiological conditions, native NSCs in the adult mouse DG generate new neurons and astrocytes, which integrate into the hippocampus' complex cellular architecture and functional circuits, and contribute to hippocampal-dependent learning, memory and mood regulation[14,35,55,56]. Here, we aimed to understand how TBI affects specific populations of NSC-derived cells in the DG. Using a combination of scRNA-seq, spatial transcriptomic and computational techniques, we: (1) identified distinct cellular populations in the AHN niche that belong to neuronal and astrocytic lineages; (2) observed that TBI induces changes in the relative proportion of cells in these lineages; (3) characterized changes in gene expression induced by TBI in specific NSC-derived cell populations; (4) confirmed the presence of NSC-derived neuronal and astrocytic cells in different subregions of the DG in situ; (5) identified a specific effect of TBI on the anatomical location of NSC-derived cell populations within the DG. Taken together, our results using a variety of methods, all support the conclusion that TBI modifies NSC fate to promote neurogenesis at the expense of astrogliogenesis, whilst also affecting the anatomical location of specific NSC-derived populations within the DG.

In agreement with observations made in other rodent TBI models, we observed an impairment in hippocampal-dependent cognitive

tasks 15 days after controlled cortical impact, concomitant with hippocampal astrogliosis, indicated by increased GFAP expression and an increase in immature neurons (DCX+ cells)[57–61]. New neurons born after TBI presented morphological alterations in the DG, including changes in dendritic shape and numbers of dendritic spines, indicating changes in neuronal integration[62,63]. TBI-induced changes in the integration of newborn neurons corresponded with increased NSC proliferation in the DG, increased progenitor and immature neuron numbers, and impairment in spatial memory.

In an attempt to understand the molecular changes elicited by moderate TBI in the AHN niche, we performed scRNA-seq, as this has proved a powerful tool to molecularly profile (rare) cell populations and how they change over developmental time, or in response to injury[64]. We performed these experiments using GFP⁺ cells from Nestin-GFP mice, which express the fluorescent marker in NSCs, their progeny, and several other cell types within the niche[25,28]. Using previously reported cell-type markers, we were able to identify 10 cell clusters representing the major cell types expected in the DG: NSCs, neuronal and astrocytic cells, oligodendrocyte precursors and other less abundant cell populations, in agreement with previous studies[27,28]. Following this first analysis, extraction of progenitors, astrocyte-like and neuron-like cells, followed by reclustering and pseudotime-based lineage tracing[65], identified two specific cell lineages within the annotated cell populations. Six neuronal cell populations (N-stage 1–6) fitted a lineage trajectory that initiated from three NSC populations (RG-like, NSC-stage 1 and 2), strongly suggesting they derive from them. An additional neuronal cell population (N-stage 7) did not fit within this lineage, suggesting a different origin, compatible with its lack of *Prox1* expression, a marker of granule cell identity in the DG[66]. Interestingly, four astrocytic cell populations (A-stage 1–4) fitted in a separate lineage trajectory, which also included RG-like, NSC-1 and NSC-2 cells, indicating they also likely originate from these NSCs. In addition, we found three astrocytic cell populations (A-stage 5–7) that did not fit within this trajectory, similarly suggesting a different origin. The three potential NSC populations we observed expressed *Ascl1, Vim, Hes5,* and *Id4*, although the levels of these four markers were lower in NSC-stage 1 cells. Indeed, recent studies have identified populations of DG NSCs characterized by the differential expression of the pro-activation factor *Ascl1* and the inhibitor of DNA binding protein *Id4*[29,35,67], indicating our NSC classification is consistent with earlier findings. This view is further reinforced both by Gene Ontology analysis, which confirmed that our NSC populations show high proliferative potential, and RNA velocity analysis which confirmed that the neuronal and astrocytic lineages described by our sequencing data are transcriptionally derived from these NSC populations.

Interestingly, in a previous study[37] Batiuk, Martirosyan, and colleagues identified two putative astrocyte populations, AST4 and AST5, which they proposed to represent a population of hippocampal NSCs or progenitor cells and a rare intermediate progenitor, respectively. This is consistent with the location of AST4 predominantly to the subgranular zone of the hippocampus and AST5 distributed between both cortex and hippocampus. Although it is difficult to compare cell types identified across different scRNA-seq studies, due to technical differences in cell-type isolation, we attempted to make a comparison of our data with their study using an integrated analysis of hierarchical clustering, pseudotime profiles and GO functional annotations. This analysis revealed that AST4 clustered closely with RGL, NSC-stage-1 and NSC-stage-2 cells with pseudotime lineage analysis revealing a distribution of AST4 cells along this developmental trajectory. Furthermore, GO analysis of the genes expressed in AST4 revealed several biological pathways associated with translation as also overrepresented in RGL, NSC-stage1 and NSC-stage2. Therefore, our analysis strongly supports the hypothesis that AST4 represents hippocampal NSCs. Despite this strong concordance between studies, we found that the RGL, NSC-stage-1 and NSC-stage-2 populations also

express the proliferation marker *Mki67*. Therefore, we consider these cells to be actively proliferative at the time animals were sacrificed, consistent with the specific capture of cells expressing GFP at high levels, which were previously shown to be proliferative stem cells[25]. In contrast, Batiuk, Martirosyan and colleagues reported that genes involved in mitosis and cell cycle control were overrepresented in AST4, suggesting some NSC astrocytic populations may retain proliferative potential in the absence of active proliferation. Taking into account the differences between the two studies, we consider the most parsimonious explanation for this discrepancy lies in the methods used for cell isolation. While the current study is biased towards the recovery of actively proliferating stem cells, Batiuk, Martirosyan and colleagues immunoisolated cells by targeting the astrocyte-specific cell surface marker ATP1B2[53]. This protein appears largely absent, or expressed at low levels, in RGL, NSC-stage-1 and NSC-stage-2 cells, consistent with the low number of AST4 cells recovered in the original study. When combined with the fact that most NSCs in adult mice are not cycling[68], we find it highly likely that few actively proliferating NSCs were sequenced by Batiuk, Martirosyan and colleagues, which, when combined with the data filtering/thresholding common in single-cell data processing, would explain the relative lack of proliferation markers, such as *Mki67*, assigned to AST4.

Crucially, we were able to confirm the sequencing data using RNAscope in situ hybridization to detect NSC-derived cell populations. For our initial assessment, we focused on cells of the astrocytic lineage, as they represent a minority of NSC progeny, and are far less characterized in the literature than their neuronal counterparts[14,55]. Specifically, we validated the presence of A-stage 1, 2, and 4 cells. We observed A-stage-1 and 2 cells to be present mainly in the SGZ, an acknowledged anatomical location for NSCs and their immature progeny in physiological conditions[14,69].

Our data support the conclusion that TBI produces no change in the developmental linages of NSC-derived neuron and astrocytes. This is in direct contrast to reports of unique cell populations associated with the development of neurodegenerative diseases such as Alzheimer's[70] and Huntington's[71], including the appearance of multiple populations of reactive astrocytes characterized by GFAP upregulation. Although these are chronic neurodegenerative diseases in which changes in cell identity may well occur over years in a non-synchronous manner, large changes in cell identity were also reported on the shorter time-scale of experimental autoimmune encephalomyelitis induction in mice[72], suggesting that responses to injury and disease development depend on a combination of initial cellular identity, cell-autonomous injury response and insult-mediated changes in the tissue (micro)environment[73]. Indeed, TBI-induced changes in NSCs appear subtle, with the major effects appearing to be on the relative abundance of NSC-derived neuronal and astrocytic cells. We found that TBI resulted in an increase in the abundance of NSC-derived neuronal populations N-stage 1–5 cells. Concomitantly, TBI resulted in a decreased abundance of the NSC-derived astrocytic populations A-stage 1, 2, and 4, without seeming to have any effect on the abundance of A-stage 3 cells. These observations suggest a concerted effect of TBI on NSC-derived cell populations, possibly promoting neurogenesis at the cost of astrogliogenesis. Supporting this conclusion, RNA velocity analysis indicates that TBI likely alters the fate of RG-like cells, increasing their probability to transition to N-stage 1 cells and decreasing their probability to transition to A-stage 1 cells. Interestingly, NSC-stage 2 cells also showed an increased probability of transitioning to N-stage 1, while NSC-stage 1 cells showed an increased probability of transitioning to NSC-stage 2 cells. These observations are consistent with an increase in neurogenesis in response to TBI. Further supporting this conclusion, N-stage 1 cells showed an increased probability of transitioning to more differentiated neuronal states−specifically N-stage 2. As N-stage 2 cells express neuroblast markers, such as DCX, this observation is compatible with an increase

in immature neuronal cells induced by TBI in the DG, which we also observe using immunohistochemical detection of DCX. Overall, RNA velocity predicts an increased progression of progenitors along the neurogenic lineage at the apparent expense of astrogliogenesis following TBI. In addition, RNA velocity predicted "back flow" transitions amongst cells in the astrocytic lineage, irrespective of experimental groups. Interestingly, several studies have showed astrocyte dedifferentiation towards immature cells with NSC potential as a response to gliosis, inflammation and injury conditions, and surgery and anesthesia[74–76]. These studies demonstrate a tendency of injury and disease to induce (mature) astrocytes to proliferate and dedifferentiate, acquiring certain properties of progenitor cells, while remaining in the glial cell lineage[73,77]. Whether this response is initiated in animals receiving control craniotomies as a response to the anesthesia and/or surgery[76], or the dissociation procedure used to produce the single-cell suspension[78] is unclear at present. Crucially, however, TBI does produce a detectable and distinct response: by favoring the A-stage 4 to A-stage 3 transition, at the expense of A-stage 3 and A-stage 4 transitions to A-stage 2, TBI appears to buffer the overall astrocytic loss across the lineage for this particular population, consistent with our observations that A-stage 3 was the only cell population not numerically affected by TBI, and our DEG analysis indicating that TBI affected the differentiation status of this population. This is in line with injury and disease-producing unique responses in astrocytes[73]. However, our in silico predictions require experimental confirmation and follow up to understand their biological relevance.

NSC proliferation and neurogenesis, induced by physiological stimuli such as running, are uncoupled from NSC-derived astrogliogenesis[14]. According to this model, TBI would be predicted to increase neurogenesis without affecting astrogliogenesis. However, contrary to this prediction, our results favor a model where NSC-derived astrogenesis is disfavored following moderate TBI, and increased astrogliosis in the DG results from the local proliferation of mature astrocytes, indicating that these two processes are differentially regulated. We can only speculate about the consequences of these changes for local tissue homeostasis at the AHN niche, but the DG is populated by multiple astrocyte subclasses, showing heterogeneity in molecular, morphological, and physiological properties. Importantly, astrocyte populations reside in distinct areas of the DG, indicating location-specific functions and interactions with local cellular environments, suggesting that subtle changes in the number and location of astrocytes likely impact DG function[79].

Our analyses of DEGs in specific cell populations indicate that A-stage 3 is the population most affected by TBI in terms of changes in cell function. In terms of individual genes affected by TBI *Ppp1r14b* is upregulated following TBI in NSCs (RG-like, NSC-stage 1, NSC-stage 2), astrocytic (A-stage 3), and neuronal (N-stage 1–4) cell populations. *Ppp1r14b* is a gene associated with proliferation and migration in other cell types[80]. *Ppp1r14b* is upregulated in hippocampal NSCs after kainic acid administration, an experimental condition that induces pathological NSC proliferation, and *Ppp1r14b* is a validated target of microRNA-137, which prevents the kainic acid-induced loss of RG-like NSCs associated with hyperproliferation in the DG[53,81,82]. Taken together, these observations suggest that *Ppp1r14b*-mediated pathways may be involved in pathological activation of NSCs and migration of some of the populations that derive from them. However, the role of *Ppp1r14b* and many other of the DEGs in hippocampal NSCs and their progeny remains unknown, warranting future studies to characterize its function(s) in detail in the context of TBI. Overall, we have predicted cellular functions that may have been affected by TBI, based on DEG analysis. These predictions require further validation, possibly by examining protein levels and investigating gene functions in specific cell types using Cre/lox-based approaches.

To better understand the effects of TBI on the spatial organization of identified cell types within the DG, we turned to spatial transcriptomics[83]. Using Molecular Cartography and a multimodal reference mapping of our scRNA-seq dataset, we were able to localize 17 cell populations that were initially annotated from the scRNA-seq dataset, to their location within the DG. Focusing on 13 populations, which represented NSCs and their progeny, we assigned cell populations to different anatomical locations in the AHN niche, and found that N-stage 3 and A-stage 1 cells, which represent two early cell populations from the NSC-derived neuronal and astrocytic lineages respectively, were misplaced in the DG of mice subjected to TBI. The alterations in the anatomical location of N-stage 3 cells is compatible with the increase in DCX+ cells observed with immunohistochemistry following TBI, and supported by our RNA velocity analysis. Changes in immature neuronal cell location impacts on their circuit integration, as they need to be physically adjacent to coordinate their lateral migration[49]. This suggests that N-stage 3 cells are related to the ectopic neurons that have been observed after TBI in the DG[84]. Although previous studies have shown that astrocytic cells are generated in low numbers from NSCs in the AHN niche[14,35], molecularly distinct, spatially organized astrocyte populations appear to support local functions in the hippocampus[37,79,85].

Recent observations have shown that physiological stimuli, such as running, stimulate adult NSC-derived neurogenesis without affecting astrogliogenesis[14]. In contrast, we show here that pathological insults, such as TBI, may stimulate adult NSC-derived neurogenesis at the expense of astrogliogenesis, highlighting the importance of assessing the balance of these processes in the AHN following pathological insults. In particular, our studies using RNA velocity and Molecular Cartography identified a transition limited progression through A-stage 1 as a major effect of TBI on NSC-derived astrocyte maturation in the DG. Astrocytes are key regulators of cell-cell interaction in the hippocampus, supporting critical aspects of circuit function from initial synapse assembly and pruning, through to control of local homeostasis and modulation of synaptic transmission[86]. In particular, astrocyte dysfunction following TBI affects the metabolic support of neurons[87] and hilar astrocytes contribute GABAergic inhibition of hippocampal dentate granule cells[88]. Moreover, neuronal rearrangements take place in the hilus after TBI, where mossy cells are sensitive and become hyperexcitable following injury[89,90]. Although it is tempting to speculate that the numerical and spatial changes in specific NSC-derived astrocytic populations may contribute to deficient metabolic support in the DG after TBI, future studies should clarify the functional relevance of these cell subpopulations. Our observation of hilar astrocytosis by immunohistochemistry and spatial transcriptomic detection of GFAP raises the interesting question how this could be compatible with decreased NSC-derived astrogliogenesis. Recent observations have demonstrated that local proliferation of mature astrocytes is a central contributor to lifelong astrogliogenesis in the dentate gyrus[14,91]. Such phenomenon could explain hilar astrocytosis even when NSC-derived astrogliogenesis is inhibited by TBI. Indeed, we observed clusters of mature (S100B + ) proliferative astrocytes in the hilus after TBI, indicating that local astrocyte proliferation and NSC-derived astrogliogenesis can be differentially regulated by TBI, and therefore they may contribute distinctly to astrocyte dynamics and hippocampal plasticity. It is unlikely that these astrocytes arose via NSC-derived astrogliogenesis, because they did not express Nestin-GFP and it takes approximately a month for NSCs to differentiate into S100B-expressing astrocytes[13]. Hence, our data indicate that the local proliferation of mature astrocytes and NSC-derived astrogliogenesis may contribute distinctly to astrocyte dynamics and hippocampal plasticity.

Overall, based on the results we describe, we propose a model for the effects of TBI on NSCs and the cell populations that derive from them in the AHN, which incorporates our key findings of changes in cell fate specification and differentiation and cell position (Fig. 5q, r).

Arguably, our work has some technical and conceptual limitations. Previous studies on the response of NSCs in the DG to TBI have delivered inconsistent observations regarding the degree to which neurogenesis is modified[24]. However, the direction of the change is disputed[60], with some studies reporting a decrease[92,93] and other studies an increase[94–96]. These discrepancies may be explained by variations in the use of different TBI models and experimental design. A previous study using CCI to investigate the effect of injury severity, concluded that moderate TBI promoted NSC proliferation without increasing neurogenesis, as measured by the number of DCX+ cells in the DG two weeks after injury[24]. In contrast, we observed a significant increase in DCX+ cells, although we used different injury induction parameters. According to a recent effort to standardize CCI parameters across different laboratories, the TBI that we induced can be defined as moderate[21]. This definition is compatible with our observation of an impairment in hippocampal-dependent learning in the Morris water maze, concomitant with an increase in neurogenesis[21,24] and defects in cell positioning and circuit integration. Regarding possible implications for TBI in humans, CCI only mimics certain aspects of human TBI. Specifically, mild, moderate or severe TBI are clinically defined in humans based on loss of consciousness, alterations in mental states, post-traumatic amnesia or coma at different times post-TBI, some of which are not considered in rodent models[21].

While the presence of adult hippocampal neurogenesis in humans has been recently debated[97,98], its impairment in older adults may play a role in neurodegenerative diseases, such as Alzheimers'[99–102]. An actionable roadmap towards a better understanding of the role of AHN in Alzheimers', and possibly other neurodegenerative diseases associated with TBI such as Chronic Traumatic Encephalopathy, has been recently proposed[103]. Importantly, outside of the hippocampus, TBI induces the expression of NSC markers in individual cells of the perilesional human cortex, suggesting that TBI may induce neurogenesis in the human brain[104].

In conclusion, the molecular and cellular changes we describe here in mouse may well help us to better understand the changes induced by TBI in the hippocampus of TBI patients, possibly opening up new targets for diagnosis, prevention and treatment of neurodegenerative diseases.

# Methods

## Animals
All animal procedures were approved by the commission for Animal Welfare at the University of Amsterdam (animal protocol CCD 4925) and/or KU Leuven (animal protocol 082/2018) and were performed according to the guidelines and regulations of the European Union for the use of animals for scientific purposes and the ARRIVE guidelines for reporting animal research[105]. Eight-week-old male Nestin-GFP$^{+/-}$ male mice[25] (available from the Jackson Laboratory as STOCK Tg(Nes-EGFP)33Enik/J), or C57/Bl6J (wild-type) male mice were used in our experiments. All mice were bred in house and housed in groups of 3–4 animals per cage throughout the experiment under a 12-h light/dark cycle (lights on at 08.00 AM) in a temperature and humidity-controlled room (21 °C, 50%) with *ad libitum* access to food and water. All mice were randomly assigned to experimental groups, and the sample size was based on previous work[20,106]. We did not consider sex in our study design and deliberately have used male mice only as sex and gonadal hormones have been shown before to influence astrogliosis after TBI in mice, indicating that the inclusion of both sexes in experimental groups could increase experimental variability[107,108]. We do not we make any claims as to whether our results are equally applicable to females.

## Controlled cortical impact
A controlled cortical impact model was used to induce a traumatic brain injury[20]. In brief, mice were placed in a stereotaxic frame and anesthetized using 2% isoflurane throughout the surgery. A craniotomy was performed, creating a 4 mm window along the skull sutures from bregma to lambda (2.5 mm posterior to and 2.5 mm lateral to the midline suture) over the left hemisphere. The skull cap was removed without damaging the dura. A 3 mm rounded stainless-steel impact piston was placed at an ~20-degree angle directly in contact with the surface of the dura, as determined using the zero-point contact sensor and a controlled impact was delivered using a Leica Impact One™ Stereotaxic Impactor (Meyer Instruments) with the following settings: 1 mm impact depth, 5.50 m/s velocity, dwell time 300 ms. After the impact, the skull bone was placed back and glued in place using medical-grade Superglue. The skin was stitched to close the wound and animals were allowed to recover on a 37 °C heat pad until fully awake.

## Behavioral testing
The effects of TBI on spatial learning were assessed 15 days after CCI using the Morris water maze test[109]. Briefly, the acquisition phase consisted of two training trials per day with an inter-trial-interval of 10 min during five consecutive days, starting at day 15 post CCI. During the acquisition phase, the platform was hidden in the NW-quadrant of the pool one cm below the water surface. On day seven, the platform was removed from the pool for a single probe trial, in which the time spent in the target quadrant was recorded. Behavior was recorded by a video camera connected to a computer with Ethovision software (Noldus, The Netherlands) for analysis, and also manually scored by an experimenter that was blind to the condition of the animals. Escape latency to platform and time spent in the target quadrant were quantified.

## Euthanasia and tissue extraction
Mice were sacrificed 15 days post-TBI or control craniotomy. Mice destined for immunohistochemistry were sacrificed by an overdose of Euthasol (pentobarbital sodium and phenytoin sodium), followed by intracardial perfusion fixation with ice-cold PBS, followed by 4% PFA. The brains were isolated and stored in PBS until further use. Mice destined for single-cell experiments were sacrificed by rapid decapitation, after which brains were removed and the dentate gyri microdissected ready for dissociation. Mice destined for experiments using either RNAscope or Molecular Cartography were sacrificed by rapid decapitation, after which the brains were removed and directly processed, according to the manufacturer's protocols.

## 5-Bromo-2'-deoxyuridine (BrdU) administration
BrdU administration was performed after CCI or control craniotomy[81]. BrdU (Sigma, St Louis, MO, USA, cat #B5002-1G) was diluted in sterile saline and administered through intraperitoneal injections at 150 mg/kg concentration (15 mg/ml of BrdU was diluted in sterile phosphate-buffered saline (PBS) with 0.01 N sodium hydroxide (1% of the total solution). Mice received 4 injections at 16, 24, 40, and 48 h after the TBI control craniotomy. Mice were sacrificed 15 days post-surgery.

## Retrovirus production and retrovirus-GFP labeling of newborn cells
Newborn cells in the DG were labeled using a retroviral vector (RV-GFP) injections[26]. HEK293T cells (CRL-3216 cell line obtained from the American Type Culture Collection, tested for mycoplasma infection using MycoAlert® Mycoplasma Detection Kit, LONZA LT07-118) were co-transfected with pCAG-GFP, pCMV-GP, and pCMV-VSV-G plasmids using calcium-phosphate precipitation in a 3:2:1 ratio. The media containing retrovirus was collected 48 h after transfection. Cell debris was removed from the supernatant by centrifugation at 3200×g for 10 min and filtration through a 0.22-μm filter (MCE membrane filter Millipore, Sigma-Aldrich). The retrovirus was concentrated by ultracentrifugation at 160,000 × g for 2 h (Sorvall WX Ultracentrifuge and

SureSpin 630 swinging bucket rotor; Thermo Fisher Scientific, Waltham, MA, USA). The retroviral pellet was resuspended in 200 μl phosphate-buffered saline (PBS; Sigma-Aldrich, St. Louis, MO, USA), aliquoted and stored at −80 °C. The titer was at $7.0 \times 10^9$ particles/ml. In total, 400 nl of this RV-GFP suspension were stereotactically injected into the DG of wild-type mice (anterior-posterior: −1.7 mm, latero-lateral: −1.6 mm, dorsal-ventral: −1.9) immediately before TBI or Control craniotomy. Following surgery, mice were placed on a warm pad and allowed to recover. Mice were sacrificed 15 days post-injection. GFP signal was detected in tissue sections using an anti-GFP antibody.

## Immunohistochemistry and image acquisition for BrdU and RV-GFP quantification

Experiments were performed following methods optimized for use in transgenic mice[11,13,110]. Animals were deeply anesthetized using 2% isoflurane and were subjected to transcardial perfusion with 25 ml of PBS followed by 30 ml of 4% (w/v) paraformaldehyde in PBS, pH 7.4. The brains were removed and post-fixed, with the same fixative solution, for 3 h at room temperature, then transferred to PBS and kept at 4 °C. Quantitative analysis of cell populations in transgenic mice was performed by means of design-based (assumption-free, unbiased) stereology using a modified optical fractionator sampling scheme as previously described[110]. Slices were collected using systematic-random sampling. The right hemisphere was selected per animal. The hemisphere was sliced sagittally in a lateral-to-medial direction, from the beginning of the lateral ventricle to the middle line, thus including the entire DG. In all, 50-μm slices (cut using a Leica VT 1200 S vibrating blade microtome, Leica Microsystems GmbH, Wetzlar, Germany) were collected in six parallel sets, each set consisting of 12 slices, each slice 300 μm apart from the next. The sections were incubated with blocking and permeabilization solution (PBS containing 0.25% Triton X-100 and 3% BSA) for 3 h at room temperature, and then incubated overnight with the primary antibodies (diluted in the same solution) at 4 °C. After thorough washing with PBS, the sections were incubated with fluorochrome-conjugated secondary antibodies diluted in the blocking and permeabilization solution for 3 h at room temperature. After washing with PBS, the sections were mounted on gelatin-coated slides with DakoCytomation Fluorescent Mounting Medium (DakoCytomation, Carpinteria, CA). Sections used for the analysis of BrdU incorporation were treated, before the immunostaining procedure, with 2 N HCl for 20 min at 37 °C, rinsed with PBS, incubated with 0.1 M sodium tetraborate for 10 min at room temperature, and then rinsed with PBS. The GFP signal from Nestin-GFP was detected with an antibody against GFP for enhancement and better visualization. The following antibodies were used: chicken anti-GFP (Aves Laboratories, GFP-1020, 1:1000 dilution); goat anti-GFAP (Abcam Ab53554, 1:1,000 dilution); rabbit anti-S100β (Dako GA50461-2, 1:750 dilution); rat anti-BrdU (Abcam Ab6326 [BU1/75 (ICR1)], 1:1000 dilution), goat anti-DCX (Santa Cruz, cat# sc-8066, 1:500 dilution); rabbit anti-Ki67 (Abcam Ab16667, 1:1000 dilution); AlexaFluor 488 donkey anti-chicken (Molecular Probes, Thermo Fisher Scientific/Invitrogen, cat # A78948, 1:500 dilution); AlexaFluor 568 donkey anti-rat (Molecular Probes, Thermo Fisher Scientific/Invitrogen, cat # A78946, 1:500 dilution); AlexaFluor 647 donkey anti-rabbit (Molecular Probes, Thermo Fisher Scientific/Invitrogen, cat # A-31573, 1:500 dilution); AlexaFluor 647 donkey anti-goat (Molecular Probes, Thermo Fisher Scientific/Invitrogen, cat # A-21447, 1:500 dilution), and DAPI (Sigma, 1:1000 dilution) was used as a counterstain when required. BrdU-positive cells were categorized with different combinations of antibodies (Nestin-GFP/GFAP; Nestin-GFP/S100β; Nestin-GFP/Mik67). All fluorescence immunostaining images were collected employing a Leica SP8 laser-scanning microscope (Leica, Wetzlar, Germany) and a 63× objective (Leica) using the manufacturer's software (Leica Application Software X) following protocols optimized for stereotaxic quantification and quantitative image analysis[13,81]. The signal from each

fluorochrome was collected sequentially, and controls with sections stained with single fluorochromes were performed to confirm antibody penetration and to exclude the possibility of fluorescence bleedthrough between acquisition channels. Images of sections from TBI and Control mice were acquired using identical settings and conditions. All images were imported into Adobe Photoshop 7.0 (Adobe Systems Incorporated, San Jose, CA) in tiff format. Brightness, contrast, and background were adjusted equally for the entire image using the "levels" controls from the "image/adjustment" set of options without any further modification. Al images shown are projections from z-stacks ranging from 10 μm (typically for imaging individual cells) through to 20 μm thickness.

## Newborn neuron morphology analysis

The dendritic arbors of RV-GFP-labeled newborn neurons were analyzed using confocal Z-stack images of GFP + /DCX+ cells in the DG[26,111]. Briefly, the number of primary dendrites emerging from the neuron soma or the presence of a main dendrite bifurcating within the first 10 μm from the soma were measured, alongside the spine density per 10-μm dendritic segment in GFP+ secondary or tertiary dendrites.

## General immunohistochemistry procedure

PFA-fixed brains were cryoprotected using 30% sucrose and then sliced into 40 μm-thick slices: every 8th section was taken for immunostaining, ensuring a 280 nm separation between slices[26]. Fluorescence immunohistochemistry was performed following a standard procedure. Sections were first incubated with blocking and permeabilization buffer (1× PBS/5% normal goat serum (Cell Signaling, cat #5425)/0.3% Triton X-100) for 30 min, followed by incubation with the primary antibody overnight at 4 °C. Sections were thoroughly washed with PBS and subsequently incubated with fluorescent secondary antibodies for 2 h at room temperature. After thorough washing with PBS, tissue slices were mounted on glass slides and counterstained with Vectashield antifade mounting medium containing DAPI (Vector Laboratories, cat# H-1200-10). The following antibodies were used: rabbit anti-GFAP (Dako, cat# Z0334, 1:10,000 dilution;) in combination with goat anti-rabbit Alexa 568 (Molecular Probes, Thermo Fisher Scientific/Invitrogen, cat # A-1101, 1:500 dilution), goat anti-DCX (Santa Cruz, cat# sc-8066, 1:500 dilution) in combination with donkey anti-goat Alexa 488 (Molecular Probes, Thermo Fisher/Invitrogen, cat # A-11055, 1:500 dilution). DAB-based immunohistochemistry for DCX was performed according to a standard protocol, as previously described[111], using goat anti-DCX (Santa Cruz, cat# sc-8066, 1:800 dilution) in combination with donkey anti-goat-biotin (Jackson Immuno Research, cat# 705-065-147, 1:500 dilution). Nissl staining was performed to counterstain nuclei using Cresyl Violet[26]. Total DCX+ cell numbers were assessed in wild-type (non-transgenic) animals using a stereological approach[112]. Gliosis was analyzed by measuring GFAP surface area coverage in the whole hippocampus using ImageJ[53].

## RNAscope® fluorescent multiplex in situ hybridization

Based on population-specific RNA-seq gene lists (Supplementary Data 3), we selected seven target transcripts in a way that their combination would allow us to distinguish cell population along the astrogenic lineage (Supplementary Fig. 7), despite the limited multiplexing capacity of the RNAscope technique[37]. This resulted in an inclusion/exclusion strategy in which the detection of some of the probe targets together with the lack of detection of others was unique to a cell type. According to this strategy, we defined A-stage 1−4 cells as follows:

A-stage 1 cells (*Slc1a3+, Hapln1−, Neat1+, Sned1+, Sparc+, Ascl1+, Frzb+*)

A-stage 2 cells (*Slc1a3+, Hapln1−, Neat1+, Sned1+, Sparc−/+, Ascl1−, Frzb−*)

A-stage 3 cells (*Slc1a3+*, *Hapln1+*, *Neat1−/+*, *Sned1−*, *Sparc+*, *Ascl1−*, *Frzb−*)

A-stage 4 cells (*Slc1a3+*, *Hapln1+*, *Neat1+*, *Sned1+*, *Sparc−*, *Ascl1−*, *Frzb−*)

where (−) indicates no expression, (+) indicates strong expression, and (−/+) indicates weak expression. Freshly removed brains from C57/Bl6J mice were directly snap-frozen using liquid nitrogen. Overall, 16 μm-thick sections containing the hippocampus were cut using a cryostat and collected directly on Superfrost glass slides. The following RNAscope® probes were used:

(i) *Slc1a3*: *Mus musculus* solute carrier family 1 (glial high-affinity glutamate transporter) member 3 (*Slc1a3*) mRNA, RNAscope® HiPlex Probe - Mm-*Slc1a3* (cat# 430781-T9);

(ii) *Haplnl*: *Mus musculus* hyaluronan and proteoglycan link protein 1 (*Hapln1*) mRNA, RNAscope® HiPlex Probe - Mm-*Hapln1*-T1 (cat# 448201-T1);

(iii) *Neat1*: *Mus musculus* nuclear paraspeckle assembly transcript 1 (*Neat1*) long non-coding RNA, RNAscope® HiPlex Probe - Mm-*Neat1*-T7 (cat# 440351-T7);

(iv) *Sned1*: *Mus musculus* sushi nidogen and EGF-like domains 1 (*Sned1*) mRNA, RNAscope® HiPlex Probe - Mm-*Sned1*-T6 (cat# 418571-T6);

(v) *Sparc*: *Mus musculus* secreted acidic cysteine-rich glycoprotein (*Sparc*) transcript variant 1 mRNA, RNAscope® HiPlex Probe - Mm-*Sparc*-T5Ascl1 (cat# 466781-T5);

(vi) *Ascl1*: *Mus musculus* achaete-scute complex homolog 1 (*Drosophila*) (*Ascl1*), mRNA, RNAscope® HiPlex Probe - Mm-*Ascl1*-T3 (cat# 313291-T3);

(vii) *Frzb*: *Mus musculus* frizzled-related protein (*Frzb*), mRNA, RNAscope® HiPlex Probe - Mm-*Frzb*-T2 (cat # 404861-T2).

The RNAscope® HiPlex v2 Assay (cat# 324102) was performed following the manufacturer's instructions[113], and images were collected on a Zeiss LSM 510 confocal laser-scanning microscope 10×, air, (N.A:0.3) and 40×, water (N.A: 1.2) objectives. A 10× overview of the dorsal DG was taken first and 40X Z-stack images with 1 μm intervals were then produced and analyzed using ImageJ.

## Molecular cartography

C57/Bl6J mouse brains were processed according to the manufacturer's protocol. Briefly, the ipsilateral side of the brain was trimmed to a dimension of maximum 1 cm thickness and immersed in a proprietary fixative (Resolve BioSciences GmbH, Monheim am Rhein, Germany), after which brains were placed in a proprietary stabilization buffer. Brains blocks were then sectioned into 2-mm slices and immersed in cryo-embedding medium (Resolve BioSciences GmbH, Monheim am Rhein, Germany), followed by snap-freezing in liquid N₂. In all, 10 μm sections were generated using a cryostat. Samples processed in this way were used for highly multiplexed single molecule in situ hybridization (Molecular Cartography platform) as described in[50]. Essentially, five tissue sections from three different animals (TBI) and four tissue sections from two different animals (Control) were stained with probes targeting 93 genes defining specific stages along both the neuronal and astrocyte lineages (Supplementary data 10). Tissue sections were counterstained with DAPI to allow cell identification through nuclear segmentation. Nine tissue sections were imaged and data files generated containing the *x–y* co-ordinates of each transcript detected. These images were processed by QuPath 0.2.3 software to segment single cells from the granule layer of the dentate gyrus (DG) (subdivided in GL1 and GL2), subgranular zone (SGZ) and hilus[53], based on the DAPI signal (using QuPath's cell detection algorithm with the parameters indicated in Supplementary Data 11). Co-ordinates for segmented nuclei were transferred to ImageJ 2.0.0-rc-43/1.52n, and the transcript count per nucleus was extracted using the Polylux_V1.6.1 ImageJ plugin, developed by Resolve Biosciences.

## Preparation of single-cell suspensions

Single-cell suspensions were prepared from individual microdissected ipsilateral dentate gyri (5–6 animals per experimental condition), using a neural tissue dissociation kit (Miltenyi Biotech), according to the manufacturer's protocol. Enzymatic digestion was followed by manual dissociation using first a P1000 pipet tip, followed by further dissociation using a P200 pipet tip. After the final round of mechanical dissociation, the suspensions were filtered using a 40-μm cell strainer and collected in HBSS containing RNase inhibitor. Individual cell preparations were pooled by condition, and centrifuged at 300×$g_{Av}$ for 12 min, after which the cell pellet was resuspended in PBS containing 0.5% FBS.

## Fluorescence-activated cell sorting

GFP-positive cells were isolated from the dentate gyrus of Nestin-GFP mice using fluorescence-activated cell sorting (FACS) with a BD FACSAria™ III Cell Sorter. Propidium iodide (PI) was added to the single-cell suspension to discriminate between live and dead cells. In a first step, doublets and cell debris were removed based on forward and side scatter. Dead cells were then removed based on PI staining, after which the single-cell suspension was sorted into GFP-positive and negative populations based on the intrinsic GFP signal recorded in the FITC channel (Supplementary Fig. 1). GFP-positive populations were sorted and collected in PBS containing FBS. Cell concentration was adjusted to 1000 cells/μL. Other studies have used similar strategies in which specific NSC populations are labeled with fluorescent proteins controlled by different cell-type-specific promoters[29,114].

## Library preparation

Single-cell suspensions were prepared from microdissected DGs obtained from 8-week-old Nestin-GFP mice, using Control=5 and TBI = 6 animals per condition. Library preparations for scRNA-seq were performed using a 10X Genomics Chromium Single Cell 3' Kit, v3 (10X Genomics, Pleasanton, CA, USA). The cell count and the viability of the samples were assessed using a LUNA dual fluorescence cell counter (Logos Biosystems), and a targeted cell recovery of 6000 cells per sample was aimed for. Post-cell counting and QC, the samples were immediately loaded onto a Chromium Controller. Single-cell RNA-seq libraries were prepared according to the manufacturer's recommended protocol (Single cell 3' reagent kits v3 user guide; CG00052 Rev B), with library quality checked at all indicated protocol points using a Qubit to measure cDNA concentration (Thermo Fisher) and Bioanalyzer (Agilent) to measure cDNA quality. Single-cell libraries were sequenced on either an Illumina NovaSeq 6000 or HiSeq4000 platform using a paired-end sequencing workflow with the recommended 10X v3 read parameters (28-8-0-91 cycles). We aimed for a sequencing coverage of 50,000 reads per cell.

## SC data preprocessing and clustering analysis

Data were demultiplexed and mapped using a standard CellRanger 3.0.2 workflow making use of the UCSC mouse genome GRCm38/mm10 assembly and refdata-cellranger-mm10-3.0.0 reference dataset. Libraries having low or high RNA content were removed to exclude cells with degraded RNA or potential cell doublets, respectively. Clustering analysis was done using the R Seurat_3.2.0 package[34] using the standard workflow. *Malat1* was excluded from the analysis, since many reads mapped to it, creating an artificial peak. Datasets were normalized by the global-scaling normalization function *LogNormalize*. Variable features were found using the *FindVariableFeatures* default variance stabilizing transformation (vst) method, by fitting log(variance) and log(mean) using local polynomial regression (loess). Canonical Correlation Analysis (CCA) was performed for the integration of anchors by *FindIntegrationAnchors* on 20 CCA dimensions and *IntegrateData* on the top 20 CCAs. The integrated data was scaled and centered using the *ScaleData* function on all transcripts. Principal

Component Analysis (PCA) on the resulting data was performed using 30 dimensions. A resampling *Jackstrow* test was performed to assess the significance of PCA components. The percentage of variance explained by each PCA was saturating at number 20. Therefore, the 20 most significant PCs were selected for UMAP reduction using *RunUMAP*, *FindNeighbors* and *FindClusters* functions. Except for the number of PCA components, no other default parameters were changed in these functions. Sub-clustering was done using the same procedure, except that only cells belonging to the astrocytic and neuronal lineages (including NSC and progenitor cells) were used (taking into account the top 30 CCA and top 16 PCA dimensions). The resolution of the initial high-level cell-type clustering was set to 3.0, and when reclustering data it was set to 0.8.

## Population enrichment analysis

The significance of population enrichment by TBI or Control samples was tested with a two-sided binomial test, using the R base *binom.test* function. The probability of success was set to represent the proportion of sequenced cells originating from the TBI samples against the whole database (62.96% for the higher-order clustering (Fig. 2c) and 57.39% in the case of the pseudotime analysis (Fig. 2f).

## Differential expression analysis

After reclustering, astrocyte and neuron data went through an additional round of SCT normalization, performed using the Seurat *SCTransform* function with default parameters, to regress out noise arising from mitochondrial genes. Differential gene expression analysis to detect cell-type-specific markers (Supplementary Data 3) was performed on the SCT normalized data using the Seurat *FindMarkers* function, using a 0.25 threshold on the ln-fold expression difference and requiring a minimum of 25% of the cells in a given population to express the marker gene. The significance of each marker was calculated using a Wilcoxon rank sum test and corrected using the Bonferroni method. To detect differentially expressed genes between equivalent cell populations identified in Control and TBI conditions, MAST tests were performed (using an extension in the Seurat *FindMarkers* function) with subsequent Bonferroni correction (Supplementary Data 7). Genes were taken to be differentially expressed (DEGs) if they showed at least a 0.2 ln-fold change in a minimum of 25% of cells within a given population.

## Integration of the scRNA-seq and spatial data

Spatial and single-cell data were integrated in order to find matching populations. For this purpose, all the extracted spatial data was merged with the single-cell data, using the R Seurat_3.2.0 package anchor-integration method. Prior to integration, spatial and single-cell datasets were separately log-normalized and variable features for each were identified, using *LogNormalize* and *FindVariableFeatures* with vst methods, respectively. Next, integration anchors were found using the *FindIntegrationAnchors* function, using 8 dimensions of CCA reduction. Next, the two databases were merged using the *IntegrateData* function, using 8 CCA dimensions. Finally, the integrated data was scaled; PCA and PCA-based UMAP analysis were performed on 8 PCA dimensions using *RunPCA*, *RunUMAP*, *FindNeighbors* and *FindClusters* functions with resolution set to 1 and all other parameters set to default. Clouds representing only the spatial data were identified. These clusters are either driven by technical factors or represent mature cells (not present in the single-cell database) and were, therefore, removed before the integration method described above was repeated using the following parameters: 10 CCA, 10 PCA, and 0.8 clustering resolution.

## Pseudotime analysis

Lineages were constructed using the Monocle v3_1.0.0 R package[36], using the *cluster_cells, learn_graph,* and *order_cells* functions with default parameters, based on clustering obtained using the UMAP dimensionality reduction method and the integrated normalized expression matrix in Seurat. NSC-like populations (NSC-stage-1, NSC-stage2, and RG-like populations from the 10x scRNA dataset and clusters 7–8 from the integrated spatial and scRNA datasets) were used as the root from which developmental pathways were developed. N-stage 7 cells and A-stage 5–7 cells clustered separately in UMAP space, indicating that they likely represent non-NSC-derived lineages. Therefore, they were excluded from pseudotime analysis.

## Pathway analysis

Gene enrichment and functional annotation analyses of cell population-specific differentially expressed (up/downregulated) genes were performed using GO and KEGG databases accessed through clusterProfiler version 3.18.1[115]. First, an unbiased analysis was performed to report all the KEGG pathways/GO terms related to the gene list of interest. This was followed by sorting based on the following keywords: 'Synapse', 'Axon', 'Neuron', 'Nervous', 'Glia', 'Astrocyte', 'Microglia', 'Injury', 'Growth', 'Tnf', 'Neuro', 'Age', 'Myelin', 'Sheath', 'Reactive', 'Ion', 'Proliferation', 'Genesis', 'Development', 'Morphology', 'Formation', 'Circuit', 'Axonogenesis'. The corresponding outputs for the biased and unbiased analysis are reported in Supplementary Data 4 and 5 (for marker genes), Supplementary Data 8 and 9 (for up or downregulated genes in TBI) log fold change >0.2 and adjusted *P* value < 0.05.

## RNA velocity analysis

Analysis was performed on the R velocyto.R_0.6 package, using spliced and unspliced RNA counts obtained through the standard run10x workflow using the gRCm38/mm10 genome. UMAP embedding space was imported from the respective Seurat clustering analysis. RNA velocity was calculated for Control and TBI separately, after removing the N-stage 7 and A-stage 5, 6, and 7 populations. The following workflow was implemented:

**Step 1.** *RunVelocity* function was used with default parameters, except *spliced.average = 0.5, fit.quantile = 0.02, kCells = 20* to obtain "current" and "deltaE" matrices.

**Step 2.** The "current" and "deltaE" matrices from step 1, and the UMAP embeddings from clustering analysis, were used in the *show.velocity.on.embedding.cor* function to obtain "tp" (transition probability matrix), "cc" (velocity-correlation matrix) and RNA Velocity on embeddings. All parameters were used with default settings, except the neighborhood size which is reported in Supplementary Data 12.

**Step 3.** For every population, the mean transition probability to each of its neighbors was calculated, using the *show.velocity.on.embedding.cor* function, where *emb* represented the UMAP embeddings of the clustering analysis (Fig. 2d) and *vel* represented the list of "current" and "deltaE" matrices (step 1) for the subpopulation under consideration. The neighborhood size, *n*, was set according to the number of cells in the populations of interest, using a minimum of 300 cells. The velocity-correlation matrix, *cc*, was obtained from step 2 for the population of interest. Parameter *scale = 'sqrt'* was used. The remaining parameters were used with default settings. The "tp" matrices obtained after step 3 were used in step 4. These data are available as Supplementary Data 15.

**Step 4.** Using "tp" matrices obtained from step 3, box plots were constructed. Box plots show the median, first quartile (25%), third quartile (75%) and interquartile range. Whiskers represent data minima and maxima; dots are data points located outside the whiskers, * marks a significant difference between Control and TBI, based on independent non-parametric two-sided Wilcoxon tests.

**Hierarchical clustering, construction of pseudotime profiles, and comparative functional annotations for cell populations**

**Hierarchical clustering.** Gene lists for each cell population were obtained from each study included in the comparisons and a similarity matrix was constructed to quantitatively represent the degree of overlap between gene sets. The matrix elements were populated with the count of common genes between corresponding pairs of studies. Hierarchical clustering was employed to group studies based on the similarity matrix using the Hclust function from the stats package (version 4.1.1) on R version 4.1.1. Clusters within the dendrogram were identified using the cutree function, specifying the desired number of clusters. In this analysis, three clusters were defined.

**Pseudotime analysis.** Pseudotime trajectories were constructed using R packages Tidyverse (version 1.3.1), Ggpubr (version 0.4.0), and Ggforce (version 0.4.1) on R version 4.1.1, based on the temporal ordering of cells reported in refs. 27 and [29]. Subsequently, gene lists from ref. 37 and the current study were overlaid onto their respective pseudotime trajectories, associating gene expression profiles with temporal progression.

**DAVID functional GO analysis.** Gene enrichment and functional annotation analysis were conducted using the DAVID tool (https://david.ncifcrf.gov/home.jsp). A stringent threshold of a minimum of three genes per term and an EASE score of 0.01 were employed. Enriched Gene Ontology (GO) terms and pathways were identified, and significance was determined by a false discovery rate (FDR)-adjusted $P$ value < 0.05.

**Figure preparation**

Figures were prepared using R v3.6.0/v4.1.0, RStudio 1.0.136/1.4.1106/v4.1.2, Adobe Photoshop CC 2022 version 23.3.2, Adobe Illustrator 2024 version 28.3. GraphPad Prism 10 for macOS version 10.1.1 was used for all statistical analyses shown in Fig. 1 and Fig. 5j–o. Schemes in Fig. 5q, r were created with BioRender.com.

**Reporting summary**

Further information on research design is available in the Nature Portfolio Reporting Summary linked to this article.

## Data availability

The scRNA-seq datasets generated during the current study are available in the Gene Expression Omnibus repository, GEO accession GSE230942. Spatial transcriptomics datasets can be found at https://doi.org/10.5281/zenodo.10829090. Source data corresponding to Figs. 4c–i are available at: https://zenodo.org/records/10829090/files/AstrocyticAndNeuronalLineage_10x.RData?download=1. Source data are provided with this paper.

## Code availability

Codes are available at https://doi.org/10.5281/zenodo.10777580 and https://doi.org/10.5281/zenodo.10829090.

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

## Acknowledgements

P.B., G.M., N.R., W.K., R.A.C., I.v.d.V., E.T., I.D., S.M.-S., O.P., J.M.E., and C.P.F. were supported by the European Union's Horizon 2020 research and innovation program ERA-NET-NEURON (grant EJTC 2016) to C.P.F. and J.M.E., by the Netherlands Organization for Scientific research (NWO, cofund ERA-NET-NEURON EJTC 2016 to C.P.F.), and by Alzheimer Nederland (grant WE03202007 to C.P.F.). JME acknowledges funding from La Caixa Foundation R23-00860; MICINN with FEDER Funds PCIN-2016-128 and PID2019-104766RB; MICINN CPP2022009779 and Basque Government PIBA_2021_1_0018. S.M.-S. acknowledges funding from MICINN RYC-2021-033215-I and Basque Government PIBA_2023_1_0045. P.J.L. is supported by the Center for Urban Mental Health, University of Amsterdam, The Netherlands, and by Alzheimer Nederland. MGH's work in Leuven was supported by the Belgian Scientific Research Fund (Fonds Wetenschappelijk Onderzoek —FWO—Grants G066715N, 1523014N, and I001818N) and the Belgian Alzheimer's Society (SAO) (Grant S#16025). MGH is currently the ERA Chair (NCBio) at i3S Porto, funded by the European Commission (H2020-WIDESPREAD-2018-2020-6; NCBio; 951923). A.M. acknowledges funding from the Stichting Alzheimer Onderzoek (SAO #2020034) and VIB Tec Watch funding. F.P. was supported by a Fundação para a Ciência e a Tecnologia (FCT) Ph.D. fellowship (2020.08750.BD). The authors acknowledge the help and advice of Dr. Dirk-Jan Saaltink in the graphic design and preparation of the cell schemes in Figs. 1o and 5g, Scheme in Fig. 5g is used with his author's permission. The RV-GFP vector was kindly provided by Kristoffer Riecken at the Medical Center Hamburg-Eppendorf (UKE), Germany. We gratefully acknowledge the use and support of the confocal microscopes at The van Leeuwenhoek Centre for Advanced Microscopy, Section of Molecular Cytology, Swammerdam Institute for Life Sciences, University of Amsterdam.

## Author contributions

P.B. designed and performed in vivo animal experiments, including controlled cortical impact, behavior, tissue extraction, sample preparation for transcriptomics, immunohistochemistry and RNAscope, he analyzed and interpreted the data derived from these experiments, and assisted in drafting the manuscript and figures. G.F.M., N.R., W.K., R.A.C., I.v.d.V., E.T., O.P., and I.D. contributed to the acquisition and analysis of immunohistochemistry and RNAscope data. P.J.L. contributed to the experimental design, provided funding and corrected the manuscript. I.D. and S.M.-S. performed BrdU and RV-GFP injections, collected samples and analyzed data, J.M.E. supervised optimization of the controlled cortical impact technique, BrdU and RV-GFP injections, contributed to experimental design, conception of the experiments, data interpretation and corrected the manuscript. A.M. led all the computational analysis, except for the RNA velocity analysis, which was performed by A.A., and the comparison of gene expression profiles from astrocyte/NSCs reported in different studies, which was performed by O.P. A.M. supervised the pathway overrepresentation analysis performed by S.H. F.P., N.K., B.N., and A.B. helped design and perform spatial transcriptomics experiments. F.P., N.K., and B.N. contributed to the design of custom gene panel and sample preparation. A.B. performed the experiments. F.P. and B.N. contributed to primary analysis of the spatial transcriptomics data. S.P. provided the resources and guidance for the preparation of single-cell RNA-seq experiments and generated cDNA libraries for single-cell RNA sequencing. S.P. also assisted with the sequencing of the single-cell libraries generated in this work. T.G.B. co-supervised the computational analyses and interpretations performed by A.M., A.A., S.H., and F.P. M.G.H. provided funding, access to the single-cell sequencing facility at VIB-KU Leuven and arranged early access to the Molecular Cartography platform. M.G.H. also supervised the sequencing and spatial trancriptomics data acquisition, computational analysis and interpretation. C.P.F. conceived the experiments, provided funding, participated in the design, execution, and interpretation of the in vivo experiments, contributed to the interpretation of sequencing and spatial trancriptomics data and composed the final version of the figures. All authors contributed to the design of figures and drafting of the initial manuscript. The final version of the manuscript was written by C.P.F. and M.G.H. All authors have approved the submitted version and have agreed both to be personally accountable for the author's own contributions and to ensure that questions related to the accuracy or integrity of any part of the work, even ones in which the author was not personally involved, are appropriately investigated, resolved, and the resolution documented in the literature.

## Competing interests

M.G.H. acted as a paid consultant to Resolve Bioscience during the development of their Molecular Cartography platform. A.M. is currently a full-time employee at Muna Therapeutics. The remaining authors declare no competing interests.
