## [Peer Review File · Nature Communications]

REVIEWER COMMENTS

Reviewer #1 (Remarks to the Author):

Manuscript title: Traumatic brain injury promotes neurogenesis at the cost of astrogliogenesis in the adult hippocampus

Authors: Bielefelder et al, Fitzsimons

This paper investigates the effect of traumatic brain injury (TBI) on neurogenesis and astrogenesis by adult neural stem cells (NSCs). While the impact on adult hippocampal neurogenesis has been demonstrated already, the authors focus on the lesser understood process of astrogenesis. Here, Bielefelder et al use single cell RNA sequencing (scRNAseq) and spatial transcriptomics to describe changes in gene expression of hippocampal cell types with and without TBI.

Focusing on astrogliogenesis and comparing it to neurogenesis is an interesting and definitely underexplored angle of TBI in the hippocampus, and RNAseq data revealed novel molecular cell types within the adult hippocampal NSC lineage. Overall, the study is well structured and performed, and well discussed. The findings are interesting and the techniques used are innovative and state-of-the-art. However, there are some comments that need to be taken into account:

- The text is written succinctly, which is mostly good, however, some passages require more information and better explanations. For example, please better explain craniotomy and the use of a mild model of TBI. How is it done? What is the main affected area?
- In Fig.1a the Box indicate the impact area. Why is there no GFAP signal? I would expect reactive gliosis specifically within the primary impact zone, while the images show it only in areas that are affected secondarily (Hippocampus).
- The data show more DCX+ proliferating progenitor cells. Does this also lead to an increased production, and integration of more newborn neurons?
- While the increase in proliferating progenitor cells as well as the behavioral data appears clear, I have difficulties in correlating them. Is it likely that an increase in progenitor cells (not increase in newly generated and integrated neurons) induce such behavioral changes in the Morris water maze? This reviewer finds it hard to believe. To resolve this point, more data on neuronal maturation and integration under TBI condition are needed.
- The paragraph on scRNA-seq is convincing and interesting. Do the authors find proliferating astrocytes as described in Batiuk et al, 2020? What is meant by astrocytes "which may retain some proliferative potential"?
- In the RNA velocity analysis, I was wondering why all "arrows" from A-stage 1 and A-stage3 and 4 are conveying onto A-stage2? What could that mean?
- Even though well described, the differential gene expression analysis revealed cell type-specific responses to TBI. However, like in many other scRNA-seq paper these days, it is a mere sum up of genes. What do we gain from these informations?
- For their RNAscope analysis, the authors chose a set of 7 probes. What were the reasons to decide for those candidate genes? Please explain better.
- If I understand the description correctly, spatial transcriptomics revealed a higher percentage of A-stage1 cells in the hilus. While the authors state that this indicates an effect on the location of NSC-derived cells, what does it mean to the astrocytes in specific DG areas? Are hilus astrocytes increased in number upon TBI? If yes, how would that fit to their observed decrease of astrogenesis? This point would be well possible to confirm spatial transcriptomics by immunostainings/RNAscope in mouse sections comparing sham operated vs TBI.
- Along the same line: the last sentence is very speculative. In contrast to scRNA-seq, spatial transcriptomics did not rely on Nestin-GFP expressing cells. It may therefore well be that other astrocyte populations which were not targeted by Nestin-GFP based scRNA-seq contribute to the GFAP signal in the hilus. This should be stated more clearly in the text.
- I suggest a better labeling of graphs in the figures (especially Y axis). It is impossible to understand the graphs without the legends.

Reviewer #2 (Remarks to the Author):

The study by Bielefeld et al. presents a detailed transcriptomic study that investigates the neuro- and astrogliogenesis in the mouse dentate gyrus after TBI. The study utilizes single-cell and spatial transcriptomics to provide detailed analysis of the cellular and molecular response and its organization following the insults. The authors characterized NSC-derived populations of cells, described changes in their relative proportion and trajectories after TBI, confirmed the presence of populations by spatial transcriptomics and identified the effect of TBI on their anatomical location. Overall, the data indicates that TBI promotes neurogenesis at the cost of astrogliogenesis and affects the anatomical location of several NSC-derived populations within the dentate gyrus.

This study represents a nice example of the combination of modern techniques for transcriptomic analysis, it is well designed and performed, the methodology is sound and meets the standards in the field. The manuscript is in general well written and the conclusions supported by data. Still, there are some points, which need to be addressed.

Major points:

- I do miss verification/validation of the increased neurogenesis at the cost of astrogliogenesis in an independent cohort of animals. As it stands, all the conclusions are made from the single scRNA-seq experiment (Fig. 2-4), therefore might be biased by particular sample prep.
- Did the spatial transcriptomics yield the identical change in the rate of neuro- and astrogliogenesis? If yes, this data would enhance the conclusion and could represent the requested validation experiment. If not, the discrepancy needs to be discussed and the validation experiment performed.
- I would appreciate, if the figures would be modified in a way to become more self-explanatory (without the need to read the figure legend). For example in Fig. 1 - add label for the impact site in the A panel; D (specify dentate gyrus), H (scheme of cell structures for respective categories), I, J (scheme for behavioural test + its name + define timepoint in J) etc.

Minor points/suggestions:

- I would suggest simplifying/reducing the number of panels the Fig. 3 (also consider to place here the scheme from Fig. 6)
- I am not convinced on the importance of the results from RNA-scope in the Fig. 5 (could be in the supplement)
- for Fig. 5 -> g-h) it is little confusing to present the merged data before the individual datasets; k) would be nice to re-order (maybe also re-name) the MC clusters to place the most similar clusters on the diagonal line; n-o and q-r) I do miss the definition (or a scheme) of the SGZ and GL; scheme from Fig. 6 could become a panel in Fig. 5
- I would also welcome to provide the number of analysed animals/cells in the figure legends
- I appreciate all the data provided in Suppl. Tables, but please consider merging some of them to reduce their total number
- I would be careful to discuss the differences in the absolute number of DEGs in the chapter "Differential gene expression analysis reveals cell population-specific responses to TBI" considering the differences in the total number of cells per cluster
- for the RNA-scope, I do miss the description of the marker gene selection

Reviewer #3 (Remarks to the Author):

The authors examine the effects of brain injury on differentiation of hippocampal neural stem cells, specifically testing the hypothesis whether Nestin-pos NSCs are primed towards differentiation towards the neuronal lineage at the expense of astrogliogenesis. While the excitement for this study is somewhat limited, sophisticated molecular techniques are used – however cross-validation at protein level is lacking throughout the manuscript.

While the authors present the GFAP coverage increasing following TBI, it is very surprising that the Nestin-GFP mouse sections were not co-stained with GFAP to capture the change in the number of

GFAP cells in the SGZ – this would have been the first step in the analysis and should have been done. GFAP-positive could have been co-stained with a number of markers to validate the results.

Why were the Nestin-GFP mice not injected with BrdU to examine GFAP+/Nestin+ cells to eliminate proliferation of astrocytes as a confounding factor and thereby focus on differentiation.

Similarly, the Nestin-GFP brains have not been co-stained with the various additional markers reported in the manuscript to characterise the diversity of cells analysed with this approach and also to validate the findings. Addition of cre mouse lines would have been also helpful to validate the findings.

The difference in the number of DCX-positive cells is very small (eg. ~10%) and it is unclear what the % change is in astrocytes/mm² within the same region. On the other hand, using the scRNA approach, the neuronal lineage proportions don't appear to significantly change (Fig2C) although astrocytes lineage % does. The way the data is presented it is unclear.

There is a number of studies that report reduction in neurogenesis following TBI. In these instances, would astrogliogenesis remain unaltered or increase? The authors should have varied the impact parameters to establish how the impact force links to the outcomes including production of neurons/astrocytes and thereby boost the novelty of these findings.

It is unclear what the behavioural read out adds as it is not linked in any direct functional way with the astrogliogenesis. In addition, it contradicts a number of studies that show reduced neurogenesis and impaired cognitive performance.

It is notoriously difficult to sort for neurons via FACS, hence many groups have focused on sorting neuronal nuclei. The authors appear to have a very good recovery - have other groups published using this approach with similar success?

Response to reviewer's comments.

Point-by-point discussion.

Reviewer #1 (Remarks to the Author):

Manuscript title: Traumatic brain injury promotes neurogenesis at the cost of astrogliogenesis in the adult hippocampus.

Authors: Bielefelder et al, Fitzsimons

This paper investigates the effect of traumatic brain injury (TBI) on neurogenesis and astrogenesis by adult neural stem cells (NSCs). While the impact on adult hippocampal neurogenesis has been demonstrated already, the authors focus on the lesser understood process of astrogenesis. Here, Bielefelder et use single cell RNA sequencing (scRNAseq) and spatial transcriptomics to describe changes in gene expression of hippocampal cell types with and without TBI. Focusing on astrogliogenesis and comparing it to neurogenesis is an interesting and definitely underexplored angle of TBI in the hippocampus, and RNAseq data revealed novel molecular cell types within the adult hippocampal NSC lineage. Overall, the study is well structured and performed, and well discussed. The findings are interesting and the techniques used are innovative and state-of-the-art. However, there are some comments that need to be taken into account

A: We thank the reviewer for their positive and encouraging words regarding the quality of our work and its implications. Below we address the reviewer's comments one by one.

1. The text is written succinctly, which is mostly good, however, some passages require more information and better explanations. For example, please better explain craniotomy and the use of a mild model of TBI. How is it done? What is the main affected area?

A: Following the reviewer's request, a better explanation of the moderate TBI model and of the controlled cortical impact (CCI) procedure has been included in the introduction section (lines 77-89), while the method description has been further improved (lines 640-651). We also explain this further in our answers below. In general, we have gone through the entire manuscript and tried to identify areas where more information and/or better explanations would be beneficial and amended the text accordingly.

2. In Fig.1a the Box indicate the impact area. Why is there no GFAP signal? I would expect reactive gliosis specifically within the primary impact zone, while the images show it only in areas that are affected secondarily (Hippocampus).

A: The referee is correct, the box in Fig. 1a indicates the location of the cortical impact, which now has now been indicated with a dashed line in Fig. 1a. The lack of GFAP signal in that area indicates the loss of cortical tissue at the site of primary injury, compatible with the local impact used in this particular CCI model.

Regarding reactive gliosis, previous studies have shown that gliosis in the cortex first increases in the peri-lesion area and then rapidly decreases to pre-impact levels within the first week after CCI, which is compatible with the lack of cortical gliosis in our experiments performed 15 days post CCI. In contrast, in the characteristic absence of the initial, direct hippocampal tissue damage that is associated with a moderate TBI, gliosis in the hippocampus develops later, generally after the first week post impact, and is associated with a secondary injury phase (Susarla et al., ASN Neuro 2014; Golub & Reddy, Exp Neurol 2022). Therefore, the predominantly hippocampal gliosis that we show 15

days after CCI (Fig. 1a-c) agrees with these previous reports obtained with the CCI model of moderate TBI. This has now been clarified in the Fig. 1a and its corresponding legend (line 1075); Introduction (lines 77-89); and the results section (lines 116-118).

3. The data show more DCX+ proliferating progenitor cells. Does this also lead to an increased production, and integration of more newborn neurons?

A: To address this important point, we have performed a new series of experiments to show the increased production and integration of newborn neurons in the TBI setting.

We have traced the integration of newborn neurons in the DG using a retroviral vector expressing GFP (RV-GFP) that only labels dividing cells, which can later be visualized in hippocampal slices (van Praag H. et al., Nature 2002). The RV-GFP was injected immediately before the CCI-induced TBI or control craniotomy, and mice were sacrificed 15 days post-injection. We observed a significant increase in the percentage of newborn neurons (% of GFP+ cells that express DCX as a fraction of the total GFP+ cells) in the TBI group compared to Controls.

The persistence of retrovirally-labeled neurons with fully developed dendritic arborization and dendritic spines showed that indeed newborn neurons generated in excess integrate into the hippocampal circuit, thereby answering the reviewer's question. However, we observed morphological changes in newborn neurons in the TBI group, in addition to their numerical increase. Specifically, we observed that a significantly higher percentage of the newborn neurons in the TBI group presented dendritic abnormalities, such as two primary dendrites emerging directly from the soma, a main dendrite bifurcating within the first 10 um from the soma, or a significantly lower number of dendritic spines per distance unit. As dendritic spines are considered basic functional units of neuronal integration and their appearance typically characterizes the final developmental stages of adult-born granule neurons (Yuste et al., Nature, 1995; Zhao et al., J Neurosci 2006), we interpret these observations as supporting the conclusion that TBI affects the integration of newborn neurons in the DG.

These new results answer the reviewer's question and are now presented in Fig. 1s-v and described in the results section (lines 135-147). In addition, the methods section (lines 680-749), the legend for Fig. 1 (lines 1114-1125) and the discussion section (lines 405-409) have been modified accordingly to highlight and discuss these new data.

4. While the increase in proliferating progenitor cells as well as the behavioral data appears clear, I have difficulties in correlating them. Is it likely that an increase in progenitor cells (not increase in newly generated and integrated neurons) induce such behavioral changes in the Morris water maze? This reviewer finds it hard to believe. To resolve this point, more data on neuronal maturation and integration under TBI condition are needed.

A: We also addressed this question on integration using injections of a retroviral vector expressing GFP (RV-GFP) to specifically label newborn neurons. We now show that the increase in NSC proliferation and numbers of progenitor cells corresponds with an increased number of newborn neurons that integrate into the dentate gyrus, albeit with morphological alterations, and parallel to behavioral changes in spatial memory. This is now presented in the results (lines 142-147), Figure 1 s-v, and discussion section (lines 405-409).

5. The paragraph on scRNA-seq is convincing and interesting. Do the authors find proliferating

astrocytes as described in Batiuk et al, 2020? What is meant by astrocytes “which may retain some proliferative potential”?

A: We addressed this question experimentally, using BrdU incorporation, an immunohistochemistry-based technique commonly used to assess cell proliferation in the dentate gyrus, and with bioinformatic comparisons between some of the cell types identified in Batiuk et al, 2020 and in our current study.

Indeed, we find proliferative astrocytes in the dentate gyrus. Some of these astrocytes were mature proliferating astrocytes (S100B+/BrdU+ cells), observed as clusters within the hilus (Fig 1 g-i). Previous reports showed that S100B expression in astrocytes marks the loss of their NSC potential (Raponi et al., *Glia* 2007). This point is further explained in our answer to point 9 of this reviewer (see below). These mature local proliferating astrocytes are not likely to be generated via NSC-derived astrogliogenesis for two reasons: first, they did not express Nestin-GFP (Fig. 1 g), and second it takes approximately a month for NSCs to differentiate into S100B-expressing astrocytes (Encinas et al., *Cell Stem Cell*, 2011), while our studies took place 15 days post CCI.

In addition, we observed proliferative Nestin-GFP+/GFAP+/BrdU+ cells (Fig 1j-k), which may reflect the proliferative astrocytes with neural stem/progenitor cell characteristics described in Batiuk et al., *Nat Commun*, 2020. To answer this question in detail, we compared the astrocytic cell types we identified in our studies with those identified by Batiuk et al., *Nat Commun*, 2020. For this, we used an integrated analysis of hierarchical clustering, pseudotime profiles, and functional annotations for cell populations based on the gene expression. This analysis is now presented in a new Supplementary Fig. 4.

We included in the analysis two astrocyte populations from Batiuk et al., 2020: i.e. AST4, which was localized to the DG, and which the authors speculated to reflect hippocampal neural stem cells; and AST5, was found equally distributed between cortex and hippocampus, and which they speculated were equivalent to locally proliferating pioneer astrocytes (Ge et al., *Nature*, 2012). Our analysis revealed that AST4 clustered closely together with our RGL, NSC-stage 1 and NSC-stage 2 cells. On the other hand, AST5 clustered closely to our A-stage 6 and A-stage 7 cells, which did not fit within the astrocytic pseudotime trajectory derived from NSCs (Fig. 2 d-e), consistent with a different origin. Similarly, the pseudotime analysis in Supplementary Fig. 4 indicated an AST4 pseudotime trajectory comparable to our RGL, NSC-stage1 and NSC-stage2 cells, whereas the AST5 pseudotime trajectory was different. Finally, DAVID functional GO analysis of the genes expressed in AST4 in Supplementary Fig. 4 revealed several biological pathways associated with translation and resembling biological pathways such as “cytoplasmic translation”, “ribosome assembly” and “ribosome biogenesis” were overrepresented in our RGL, NSC-stage1 and NSC-stage2 cells. Therefore, our new bioinformatic analysis supports the original hypothesis that the AST4 population is a hippocampal neural stem cell population.

Despite this strong concordance between both studies, we found that the RGL, NSC-stage 1 and NSC-stage 2 populations we identified express the proliferation marker Mki67. Therefore, we consider these cells to be actively proliferative at the time animals were sacrificed, consistent with the specific capture of cells expressing GFP at high levels (Supplementary Fig. 1), which were previously shown to be proliferative stem cells (Mignone et al., *J Comp Neurol*, 2004). In contrast, Batiuk et al reported that genes involved in mitosis and cell cycle control were overrepresented in AST4, suggesting that some of the NSC astrocytic populations may retain proliferative potential in the absence of active proliferation, answering the second question of the reviewer.

Considering the differences between the two studies, we think that the most parsimonious explanation for this discrepancy lies in the methods used for cell isolation. While the current study is biased towards the recovery of actively proliferating stem cells, Batiuk and colleagues immunisolated cells by targeting the astrocyte-specific cell surface marker ATP1B2. This protein appears largely absent, or expressed at low levels, in RGL, NSC-stage1 and NSC-stage2 cells (Supplementary Fig. 4e), consistent with the low number of AST4 cells recovered in the original study. When combined with the fact that most NSCs in adult mice are not cycling (reviewed in Urban et al., *Neuron*, 2019), we find

it highly likely that few actively proliferating NSCs were sequenced by Batiuk et al., which, when combined with the data filtering/thresholding common in single cell data processing, would explain the relative lack of proliferation markers, such as Mki67, assigned to AST4.

These points are now presented in the results (lines 229-246), new Supplementary Figure 4, and its corresponding legend (lines 1242-1253), methods section (lines 985-1005), and included in the revised discussion (lines 437-467).

6. In the RNA velocity analysis, I was wondering why all “arrows” from A-stage 1 and A-stage3 and 4 are conveying onto A-stage2? What could that mean?

A: We do observe predicted “back flow” transitions in the astrocytic lineage (Fig. 3a, b), in both Control and TBI conditions, as indicated by the reviewer. Interestingly, several studies have revealed astrocyte dedifferentiation towards immature cells with NSC potential as a response to gliosis, inflammation, and injury conditions, and even to surgery and anesthesia. These studies demonstrate a tendency of injury and disease to activate astrocytes to proliferate and dedifferentiate thereby seemingly acquiring certain properties of progenitor cells, while remaining in the glial cell lineage (see Guo et al., Cell Stem Cell, 2014; Pestana et al., Brain Sci, 2020; and references therein). Whether this dedifferentiation response is initiated in animals receiving control craniotomies as a response to the anesthesia and/or surgery (Wang X et. al., Front Aging Neurosci, 2022), or whether it is induced by the dissociation procedure used to produce the single cell suspension (van den Brink et al., Nat Methods, 2017) is unclear at present. What is clear is that TBI produces a detectable and distinct response, increasing the probability of transitions to the A-stage 3 state, in line with our observations that A-stage 3 was the only cell population not numerically affected by TBI, and our DEG analysis indicating that TBI affected the differentiation status of the population. This is consistent with injury and/or disease producing unique responses in astrocytes (Pestana et al., Brain Sci., 2020). However, it has to be acknowledged that these *in silico* predictions require exhaustive experimental validation which is beyond the scope of the present study. We now address these issues in the results section (lines 280-285) and discussion (lines 503-519).

Finally, we wish to emphasize that, irrespective of their origin, these “back flow” transitions in no way impact the main message of the manuscript (demonstrated using a variety of independent approaches), which is that TBI increases neurogenesis at the expense of astroglialogenesis by promoting the differentiation of NSCs along the neuronal lineage, while inhibiting their transition along the astrocytic lineage.

7. Even though well described, the differential gene expression analysis revealed cell type-specific responses to TBI. However, like in many other scRNA-seq paper these days, it is a mere sum up of genes. What do we gain from these informations?

A: Differential gene expression (DEG) analysis is a commonly used approach to predict likely changes in cellular processes from scRNAseq datasets, which we used to interpret the changes in cellular processes induced by TBI in specific cell populations that were derived from hippocampal NSCs. As suggested by the reviewer, we have now added more possible interpretation to this analysis. Overall, our DEG results indicate that A-stage 3 cells are the cell population most affected by TBI in terms of changes in cellular functions. Specifically, our DEG analyses suggest that TBI affected astrocytic differentiation and promoted proliferation in A-stage 3 cells. These conclusions are supported by two observations: first, TBI upregulated a group of genes linked to neuronal specification which are commonly repressed during astrocytic differentiation (*Sox4*, *Stmn1*, *Eid1*, *Pcp4*) upregulated by TBI in A-stage 3 cells (Fig. 4a and Supplementary data table 7), and second, TBI downregulated two genes related to negative regulation of cell cycle progression (*Arhgdia*, *Nacc2*) in A-stage 3 cells (Fig 4a and Supplementary Data 7).

The conclusion that TBI promoted A-stage3 cell proliferation is also supported by the observation that A-stage 3 was the only NSC-derived population that was not depleted by TBI (Fig. 2f). However, it appears that A-stage 3 cells were not actively proliferating (i.e. expressed Mki67) at the time of sacrifice of the animals, as Mki67 expression was not detected. Furthermore, our DEG analysis identified, one gene, i.e. *Ppp1r14b*, consistently upregulated by TBI in 7 cell populations (Fig. 4a, and Supplementary data table 7). Specifically, *Ppp1r14b* is upregulated by TBI in RG-like (Fig. 4c), NSC-stage 2, (Fig. 4d), astrocytic A-stage 3 (Fig. 4e), and neuronal N-stage 1-4, (Fig. 4f-i) cell populations, suggesting *Ppp1r14b* may play an important role in the response to TBI across these cell populations. This conclusion is consistent with both the changes in NSC-proliferation (Fig 1 and Supplementary Figure 3; and Supplementary Data 3), and location of NSC-derived cells we observe (Fig. 5), as *Ppp1r14b* is a gene associated with proliferation and migration (Cao et al., Int J Oncol, 2022). However, the role of *Ppp1r14b* and other DEGs in hippocampal NSCs and their progeny remains unknown, and our observations warrant future studies to characterize their function(s) in detail, in the context of TBI.

In conclusion, in terms of gains asked by the reviewer, we have predicted cellular functions that may have been affected by TBI, based on gene-association analysis. Although these require further validation, notably in the consequences of TBI-induced changes in gene expression for relevant cellular signaling pathways, covariation data provide a valid resource for generating experimental hypotheses on gene function and regulation. These points are now included in the results (lines 289-292 and 296-307) and discussion (lines 532-534 and 544-547).

8. For their RNAscope analysis, the authors chose a set of 7 probes. What were the reasons to decide for those candidate genes? Please explain better.

A: We apologize for not being explicit enough regarding the selections of target genes. The selection of target genes studied with the RNAscope was done in a way similar to the selection of astrocyte specific genes in Batiuk et al, Nat Commun, 2020. In that paper, the authors state that “Based on closer examination of RNA-seq gene lists, a set of markers was identified to specifically label astrocytes, based on overall levels of gene expression (absent/low to high relative expression)”. Similarly, based on population specific RNA-seq gene lists (Supplementary data 3), we selected 7 target transcripts in a way that their combination would allow us to distinguish cell population along the astrogenic lineage (Supplementary Figure 7), despite the limited multiplexing capacity of the RNAscope technique. This resulted in an inclusion/exclusion strategy in which the detection of some of the probe targets together with the lack of detection of others was unique to a cell type. According to this strategy, we defined A-stage 1-4 cells as follows:

A-stage 1 cells (*Slc1a3+*, *Hapln1-*, *Neat1+*, *Sned1+*, *Sparc+*, *Ascl1+*, *Frzb+*)

A-stage 2 cells (*Slc1a3+*, *Hapln1-*, *Neat1+*, *Sned1+*, *Sparc-/+*, *Ascl1-*, *Frzb-*)

A-stage 3 cells (*Slc1a3+*, *Hapln1+*, *Neat1-/+*, *Sned1-*, *Sparc+*, *Ascl1-*, *Frzb-*)

A-stage 4 cells (*Slc1a3+*, *Hapln1+*, *Neat1+*, *Sned1+*, *Sparc-*, *Ascl1-*, *Frzb-*)

where (-) indicates no expression, (+) indicates strong expression and (-/+) indicates weak expression. According to this strategy, the description of target and probe selection has now been clarified and extended in the methods sections (lines 775-786).

9. If I understand the description correctly, spatial transcriptomics revealed a higher percentage of A-stage1 cells in the hilus. While the authors state that this indicates an effect on the location of NSC-

derived cells, what does it mean to the astrocytes in specific DG areas? Are hilus astrocytes increased in number upon TBI? If yes, how would that fit to their observed decrease of astrogenesis? This point would be well possible to confirm spatial transcriptomics by immunostainings/RNAscope in mouse sections comparing sham operated vs TBI.

A: We reasoned that if hilar astroglialogenesis is mediated through the local proliferation of mature astrocytes that are not derived from NSCs, as shown in (Schneider et al., EMBO J, 2022), such a phenomenon could explain the increase in hilar astrocytes as now shown by immunocytochemistry (Fig. 1a-c) and spatial transcriptomic (Fig. 5p), even when astroglialogenesis from NSCs is inhibited after TBI (see also the response to punt 5 of this reviewer).

To test this hypothesis, we performed an additional series of experiments in which we evaluated the effects of TBI on total cell proliferation, and on mature (S100B+) astrocyte and immature neuron (DCX+) numbers. Cell proliferation was measured using BrdU incorporation. Our results indicate that TBI increased total cell proliferation (BrdU+ cells) in the DG (Fig. 1e, f), resulting in an increase in the number of local proliferating mature astrocytes (S100B+/BrdU+ cells) (Fig. 1g-i), which were observed as clusters of S100B+/BrdU+ cells in the hilus (Fig. 1g). We further observed and increase in proliferative NSCs (Nestin-GFP+/GFAP+/BrdU+ cells) (Fig. 1j-k), with RGL morphology in some cases (Fig. 1g), and an increase in the number of proliferating immature neurons and neuroblasts (DCX+/BrdU+ cells) (Fig. 1p-r).

These new findings confirm many of the observations made using other techniques and support the interpretation that local proliferation of S100B+ (mature) astrocytes may contribute to increased astrocyte numbers, even when NSC-derived astroglialogenesis is disfavored by TBI. This indicates that these two processes can be differentially affected by TBI. These results are now described in the results (lines 118-124), legend to Fig. 1 (lines 1079-1093), methods (lines 671-677 and 696-741) and discussed in lines 521-531.

10. Along the same line: the last sentence is very speculative. In contrast to scRNA-seq, spatial transcriptomics did not rely on Nestin-GFP expressing cells. It may therefore well be that other astrocyte populations which were not targeted by Nestin-GFP based scRNA-seq contribute to the GFAP signal in the hilus. This should be stated more clearly in the text.

A: Based on the results from the experiments described in the previous point, we have deleted that sentence.

11. I suggest a better labeling of graphs in the figures (especially Y axis). It is impossible to understand the graphs without the legends.

A: Following the reviewer's request, we have re-labelled the Y axes in several figure panels, to reflect better the variables represented and how they were measured. Further, we have now included visual markers (arrows and arrowheads) and schematics indicating features of importance in some microphotographs in Fig. 1. We hope these changes make the figures clearer and more self-explanatory. Similar changes have been introduced in Fig. 2; Fig. 4; Supplementary Fig. 3; and Supplementary Fig. 7.

Reviewer #2 (Remarks to the Author):

The study by Bielefeld et al. presents a detailed transcriptomic study that investigates the neuro- and astrogliogenesis in the mouse dentate gyrus after TBI. The study utilizes single-cell and spatial transcriptomics to provide detailed analysis of the cellular and molecular response and its organization following the insults. The authors characterized NSC-derived populations of cells, described changes in their relative proportion and trajectories after TBI, confirmed the presence of populations by spatial transcriptomics and identified the effect of TBI on their anatomical location. Overall, the data indicates that TBI promotes neurogenesis at the cost of astrogliogenesis and affects the anatomical location of several NSC-derived populations within the dentate gyrus.

This study represents a nice example of the combination of modern techniques for transcriptomic analysis, it is well designed and performed, the methodology is sound and meets the standards in the field. The manuscript is in general well written and the conclusions supported by data. Still, there are some points, which need to be addressed.

A: We thank the reviewer for their positive and encouraging words regarding our work. Below we address all the points indicated by the reviewer.

Major points:

1. I do miss verification/validation of the increased neurogenesis at the cost of astrogliogenesis in an independent cohort of animals. As it stands, all the conclusions are made from the single scRNA-seq experiment (Fig. 2-4), therefore might be biased by particular sample prep.

A: Due to their similarity, we addressed this request from the reviewer together with point 9 from reviewer 1. Therefore, it is elaborated only shortly here, and we refer to our answer to point 9 from reviewer 1 for details. We validated our scRNA-seq and spatial transcriptomics experiments using a new independent cohort of animals in which we measured effects of TBI on total cell proliferation, mature (S100B+) astrocytes and immature neuron numbers (DCX+ cells). Our results indicate that TBI increased total cell proliferation (BrdU+ cells) in the DG, resulting in an increase in the number of local mature proliferating astrocytes (S100B+/BrdU+ cells), which were observed as clusters in the hilus. We also observed an increase in the number of proliferating immature neurons and neuroblasts (DCX+/BrdU+ cells) (Fig. 1 e-r).

Furthermore, in another series of experiments, we traced the integration of newborn neurons in the DG using a retroviral vector expressing GFP (RV-GFP) that only labels dividing cells, and that can be visualized in hippocampal slices (van Praag H. et al., Nature 2002). The RV-GFP was injected immediately before the CCI-induced TBI or control craniotomy, and mice were sacrificed 15 days after. We observed a significant increase in the percentage of newborn neurons (% of GFP+ cells that express DCX as a fraction of the total GFP+ cells) in the TBI group compared to Controls (Fig 1s-v).

These observations support an increase in neurogenesis, in two additional independent cohorts of animals. As mentioned above, this confirms our earlier observations and supports our interpretation that local proliferation of S100B+ (mature) astrocytes may contribute to hilar astrogliogenesis after TBI, even when NSC-derived astrogliogenesis is disfavored. The manuscript has been amended in relevant sections to describe these experiments and findings: this includes the results (lines 119-147), legend to Fig. 1 (lines 1079-1093 and 1104-1124), methods (lines 671-748) and discussion (lines 405-409 and 521-531).

2. Did the spatial transcriptomics yield the identical change in the rate of neuro- and astrogliogenesis? If yes, this data would enhance the conclusion and could represent the requested validation experiment. If not, the discrepancy needs to be discussed and the validation experiment performed.

A: To address this question, we have prepared an additional figure (Supplementary Fig. 7g) and data resource (Supplementary data 13), comparing the contribution of cells to each supplementary Molecular Cartography (MC) cluster per experimental condition, similar to what we show for the scRNA-seq data in Fig. 2f. In this new figure, we show that the number of cells in MC cluster 1, which contains N-stage 1 and 2 cells (early neuronal cells), is significantly larger than expected in the TBI group and the number of cells in MC clusters 3 and 4, which contain A-stage 2-4 cells (astrocytic cells), is significantly smaller than expected in the TBI group, thereby showing a correspondence between the scRNA-seq and spatial transcriptomics with respect to the rate of neurogenesis/astrogliogenesis. These observations are now described in the results (lines 357-365), and Supplementary Fig. 7g and its corresponding legend (lines 1273-1278).

3. I would appreciate, if the figures would be modified in a way to become more self-explanatory (without the need to read the figure legend). For example in Fig. 1 - add label for the impact site in the A panel; D (specify dentate gyrus), H (scheme of cell structures for respective categories), I, J (scheme for behavioural test + its name + define timepoint in J) etc.

A: We have addressed this request from the reviewer together with point 11 from reviewer 1, as they seemed similar. The Y axes have been re-labeled in several figure panels for greater clarity. Similar changes have been introduced in Fig. 2; Fig. 4; Supplementary Fig. 3; and new Supplementary Fig. 6. Schemes of cell structures used for classification have been included in several figures, including Fig. 1o and Fig. 5 g, h. Details regarding how the behavioral test were performed have been added to Fig. 1w, x and the methods section (lines 655-659).

Minor points/suggestions:

A: In general, we appreciate the value of these suggestions. We explain below why and how we have decided to incorporate them or not, keeping in mind the readership interest and clarity of the manuscript.

4. I would suggest simplifying/reducing the number of panels the Fig. 3 (also consider to place here the scheme from Fig. 6)

A: Following the reviewer's suggestion, we have simplified Fig. 3 and its description. The figure now consists of 7 panels containing the most crucial information needed to support the conclusions made regarding NSC fate (Fig 3 a-g). These include two overview panel with the RNA velocity trajectories for all NSC-derived neuronal and astrocytic clusters in Control and TBI conditions (Fig. 3a, b) and the mean transition probabilities calculated from RNA velocity in Control and TBI conditions for the three NSC clusters we identify by scRNA-seq (NCS-stage 1, NSC-stage 2, RG-like cells) and the populations that directly originate from them, N-stage 1 and A-stage 1 cells (Fig. 3c-g). The transition probabilities calculated for other NSC-derived neuronal and astrocytic populations are now presented in a new Supplementary Fig. 6. The results (lines 252-286), legends to Fig. 3 (lines 1155-1158) and Supplementary Fig. 6 (lines 1258-1262), have all been modified accordingly.

In addition, we have also added a paragraph which we hope clarifies how information provided in these plots should be interpreted, including references to the recent literature (La Manno et al., Nature, 2018; Bergen et al., Nat Biotechnol, 2020; Svensson et al., Moll Cell 2018; Lange et al., Nat. Methods 2022), in the results (lines 252-285). As suggested by the reviewer Fig 6 has been replaced to two panels in Fig. 5 (Fig. 5q, r), as we consider it an ideal final figure to bring together all key points of the paper, including differences in spatial location, which are only addressed in this last section.

5. I am not convinced on the importance of the results from RNA-scope in the Fig. 5 (could be in the supplement)

A: In our view, the RNA-scope data is valuable as the Molecular Cartography spatial transcriptomics technique we used in the subsequent experiments is relatively new and there are relative few reports in the literature to date (although some examples have appeared already, see text for references). We feel therefore that the inclusion of the RNA-scope dataset provides an initial validation of the observations made by scRNA seq, but then in an independent animal cohort (see also point 1 of this reviewer). Although the RNA-scope results may provide a limited conceptual advance per se, they provide technical robustness and strengthen confidence in the interesting results reported using Molecular Cartography.

We therefore kept an overview image of the RNA-scope signals in the GCL and closely adjacent tissue in the main manuscript, Fig 5a. Images reporting the signal from individual probes are now in Supplementary Fig. 7a-c. This redrafting had the additional advantage of allowing us to move final summary schematic to Fig. 5q-r, as suggested by the reviewer in point 4. We also added schemes of the SGZ/GCL to Fig 5g, h, as suggested by this reviewer (see next point 6). With these changes, we believe to have achieved a good compromise between the prominence of the RNA-scope results in the manuscript and the request from the reviewer to simplify the figures. These changes have resulted in small changes in the references to the figure panels in the results (lines 327-383), and the legends to Fig. 5a and Supplementary Fig. 7a-c (lines 1173-1213 and 1265-1278 respectively). Fig. 6 has been removed.

6. for Fig. 5 -> g-h) it is little confusing to present the merged data before the individual datasets; k) would be nice to re-order (maybe also re-name) the MC clusters to place the most similar clusters on the diagonal line; n-o and q-r) I do miss the definition (or a scheme) of the SGZ and GL; scheme from Fig. 6 could become a panel in Fig. 5

A: Following the reviewer's suggestion, we have reordered the figure. Fig. 5 panels g-h in the previous manuscript are now presented as Fig. 5 panels b-c. As suggested, the individual datasets are presented before the merged data. We have also reordered the MC clusters to place the most similar clusters in the diagonal (new Fig. 5f). Finally, we provide detailed definitions of the SGZ, GL1, GL2 and hilus that we used to quantify cell positioning. These definitions are now in the results (lines 367-372).

In addition, we have included a (new) detailed schematics illustrating these subdivisions in Fig. 5g and h. Finally, we have moved the previous summary scheme from Fig. 6 to Fig. 5q-r.

7. I would also welcome to provide the number of analysed animals/cells in the figure legends.

A: Following the reviewer's suggestion, we have provided the number of animals included in each experiment in the Figure legends and referred there to the Supplementary Data file where the number of cells per cluster and experimental condition can be found. Additionally, we have included a new Supplementary Data 13, where we report the number of cells in each Molecular Cartography cluster per experimental condition. Finally, we have changed bar plots into dot plots or box plots, so that the data distribution is more readily accessible to the reader, and we have prepared a data source file containing all our data underlying the means/averages in box plots, bar charts, and tables, as indicated in the instructions to authors.

8. I appreciate all the data provided in Suppl. Tables, but please consider merging some of them to reduce their total number.

A: All the Supplementary tables have been renamed as "Supplementary Data" as indicated in the instructions to authors and merged into a single Excel file, in which each of the Supplementary Data is now included as a separate sheet.

9. I would be careful to discuss the differences in the absolute number of DEGs in the chapter "Differential gene expression analysis reveals cell population-specific responses to TBI" considering the differences in the total number of cells per cluster

A: Following the reviewer's suggestion, we deleted this sentence.

10. for the RNA-scope, I do miss the description of the marker gene selection

A: We addressed this request from the reviewer together with point 8 from reviewer 1, as the comments seemed essentially identical to us. As such, a full description of target and probe selection is now included in the methods (lines 775-786).

Reviewer #3 (Remarks to the Author):

The authors examine the effects of brain injury on differentiation of hippocampal neural stem cells, specifically testing the hypothesis whether Nestin-pos NSCs are primed towards differentiation towards the neuronal lineage at the expense of astrogliogenesis. While the excitement for this study is somewhat limited, sophisticated molecular techniques are used – however cross-validation at protein level is lacking throughout the manuscript.

A: We thank the reviewer for their effort in reading our manuscript and expressing their opinion, but we respectfully disagree, as we consider only part of this statement to be correct. While we indeed did not validate all our transcriptomics results at the individual protein level (which would involve an enormous extra body of work), we have validated some important ones by histochemistry in our revised manuscript (see our detailed response below).

However, even if we were to extend this work and perform a much more extensive validation, we do not believe it would not help us to achieve the objectives of our study. We aimed to assess how TBI affects NSC fate in the adult hippocampus, how it changes neurogenesis and astrogliogenesis and, consequently, how TBI affects the relative cellular composition of the AHN niche. For this, we constructed a cellular catalogue of the AHN niche using scRNA-seq, which is a powerful and frequently used tool to profile, identify, classify, and discover new or rare cell types and subtypes (see Jovic et al., Clin Transl Med. 2022). As such, scRNA-seq represents a highly suitable tool to achieve our objectives, which we independently validated using highly multiplexed, single cell spatial transcriptomics, a strategy that generally provides reliable validation of scRNA-seq (Longo et al., Nature Reviews Genetics, 2021).

Additionally, based on gene-association analysis, we further predicted functions, as also requested by referee 1. Such covariation data provide a valuable resource for generating hypotheses on gene function and regulation (Chen et al., Science, Vol 348, Issue 6233, 2015). Moreover, a request to cross validate transcriptomic datasets at the protein levels immediately raises the question of which genes should be validated. The presence of only a partial correlation between RNA and protein expression is furthermore a well-known phenomenon for several genes (Payne, Trends Biochem Sci, 2015). As it is unknown *a priori* which gene transcripts and protein levels would correlate poorly, or not at all, in a given tissue under particular experimental conditions, two options arise: 1) to validate our transcriptomic observations by determining expression levels of all genes at the protein level or, 2) randomly select some genes for validation at the protein level.

Regarding the first option, if technically feasible (Xie and Ding, Adv Sci, 2022 for an extensive discussion of the current limitations and challenges), we consider it not realistic in terms of cost and workload, being effectively a whole new study. The second option could be an alternative, but how many genes need to be confirmed at the protein level to satisfy the reviewer's request? As some transcript will be concordant and others will be non-concordant with protein expression, adopting a random selection for validation seems unlikely to provide much added value (Coenye et al, Biofilm. 2021 for a detailed discussion of the topic). However, as we agree with the reviewer that this is a valid point regarding possible limitations of our work, we now amended the discussion to refer to this issue (lines 544-547).

Furthermore, we wanted to emphasize the novelty of our work to the reviewer with the following points, hoping this could increase her/his enthusiasm for our work:

1) While deregulation of hippocampal NSCs and neurogenesis have been observed after TBI, there is lack of information as to how TBI affects hippocampal astrogliogenesis. Recent studies have shown that NSC proliferation and neurogenesis, induced by physiological stimuli such as running, are uncoupled from NSC-derived astrogliogenesis (REF). Based on this observation, TBI would be predicted to increase neurogenesis without affecting astrogliogenesis. However, contrary to this prediction, we show that TBI increases neurogenesis while reducing astrogliogenesis, demonstrating for the first time that these two processes are paired in the pathological context of TBI. We have now highlighted this point in the revised discussion (lines 580-593).

2) Crucially, our data support the conclusion that TBI produces no change in the developmental lineages of NSC-derived neuron and astrocytes. This is in direct contrast to reports of unique cell populations associated with the development neurodegenerative diseases such as Alzheimer's and Huntington's, including the appearance of multiple populations of reactive astrocytes characterized by GFAP upregulation. Although these are chronic neurodegenerative diseases in which changes in cell identity may well occur over years in a non-synchronous manner, large changes in cell identity were also reported on the shorter timescale of experimental autoimmune encephalomyelitis

induction in mice, suggesting that responses to injury and disease development depend on a combination of initial cellular identity, cell autonomous injury response and insult-mediated changes in the tissue (micro)environment. Please see discussion for references, lines 475-484.

1. While the authors present the GFAP coverage increasing following TBI, it is very surprising that the Nestin-GFP mouse sections were not co-stained with GFAP to capture the change in the number of GFAP cells in the SGZ – this would have been the first step in the analysis and should have been done. GFAP-positive could have been co-stained with a number of markers to validate the results.

A: We agree this is valuable and have now included new experiments in the manuscript in which we stained hippocampal sections from Nestin-GFP mice from the Control and TBI groups with anti-GFAP and anti-Mki67 (a marker of proliferation expressed in NSC in our scRNA seq). Indeed, our new results show an increase in the number of Nestin-GFP+/GFAP+/Mki67+ cells in the TBI group, providing further validation of our scRNA-seq studies, a point also requested by the reviewer. These results are now presented in Fig. 1 (new panels 'j' and 'k'). Necessary changes to the figure legend (lines 1093-1098), methods (lines 718-729), results (lines 125-126) and discussion (lines 407-409) have also been made.

2. Why were the Nestin-GFP mice not injected with BrdU to examine GFAP+/Nestin+ cells to eliminate proliferation of astrocytes as a confounding factor and thereby focus on differentiation.

A: This point was also mentioned by others, and we have addressed the reviewer's request together with point 9 from reviewer 1 and point 1 from reviewer 2. We validated the conclusions made from the scRNA-seq and spatial transcriptomics experiments in a new independent cohort of animals by measuring the effect of TBI on total cell proliferation, and on the numbers of mature (S100B+) astrocytes and immature neurons (DCX+). Cell proliferation was measured using BrdU incorporation. Our results indicate that TBI increased total cell proliferation (BrdU+ cells) in the DG and led to an increase in the number of local, mature proliferating astrocytes (S100B+/BrdU+ cells), which were observed as clusters of S100B+/BrdU+ cells in the hilus. We also observed an increase in the number of proliferating immature neurons and neuroblasts (DCX+/BrdU+ cells).

These new findings confirm our earlier observations using other (transcriptomics) techniques. Recent evidence further showed that astroglialogenesis in the murine dentate gyrus is a dynamic process to which both the local proliferation of mature astrocytes and the proliferation of NSCs contribute (Schneider et al., EMBO J. 2022). Overall, our results support the interpretation that local proliferation of mature astrocytes may contribute to an increase in the number of hilar astrocytes seen after TBI, even when NSC-derived astroglialogenesis is disfavored. This indicates that these two processes can be differentially regulated by TBI. This is now presented in the methods (lines 671-677 and 696-741); results (lines 118-124); legend to Fig. 1 (lines 1081-1098); and discussion (lines 521-531).

3. Similarly, the Nestin-GFP brains have not been co-stained with the various additional markers reported in the manuscript to characterise the diversity of cells analysed with this approach and also

to validate the findings. Addition of cre mouse lines would have been also helpful to validate the findings.

A: We apologize, but we do not consider this remark to be very specific. As such, it is difficult for us to interpret and act on experimentally. In our manuscript, we describe tens of individual cell populations, defined by hundreds of different transcripts. Which specific markers and which cell populations is the reviewer asking us to validate specifically? Although functional validation is a valid point (see our previous discussion above), the problems we discuss with validation at the protein level, also apply here. Functional validation of a selection of genes using Cre mouse lines, as the reviewer is suggesting, will not guarantee that other genes have been correctly identified by our transcriptomics studies. Aside from the lack of specific Cre lines for some of the identified genes that may limit this strategy, addressing this request would require years of additional animal breeding and analysis. Overall, therefore, we consider it unlikely to provide much added value within the objectives of our current study. However, we do now discuss the option of functional validation using experimental strategies such as the Cre/lox system in the manuscript (lines 545-547).

4. The difference in the number of DCX-positive cells is very small (eg. ~10%) and it is unclear what the % change is in astrocytes/mm² within the same region. On the other hand, using the scRNA approach, the neuronal lineage proportions don't appear to significantly change (Fig2C) although astrocytes lineage % does. The way the data is presented it is unclear.

A: The reviewer is correct, we see a mild effect of TBI on the total number of DCX+ cells (Fig. 1l-n), which is possibly brought about by an increase in a subset of DCX+ neuroblasts (Fig. 1o). Now, with our new data, we also report an increase in the local proliferation of mature astrocytes (Fig. 1g-i), which answers the first part of the reviewer's point. Regarding the analysis of the scRNA seq data in Fig. 2c, this first clustering analysis included the total neuron population included in the scRNA seq experiment, and therefore the mild effects of TBI on a subset of cells may have been diluted in the total population. Indeed, the proportion of neuronal cells in the TBI group was larger than expected by chance 65.02% of the cells measured by RNA-seq came from the TBI group, whereas 62.96% would be expected by chance (Supplementary Data 1). However, this difference had a $p=0.08$, and therefore did not reach the threshold for significance we set for other populations ($p<0.05$) (Fig. 2c, Supplementary Data 1). In fact, our re-clustering analysis focused on NSCs and derived populations (Fig. 2d-f) shows that several, but not all, of the neuronal populations derived from NSC, are increased by TBI, indicating a cell population-specific effect that aligns with the immunohistochemistry data in Fig. 1. We have now clarified this point in the results (lines 172-179).

5. There is a number of studies that report reduction in neurogenesis following TBI. In these instances, would astroglialogenesis remain unaltered or increase? The authors should have varied the impact parameters to establish how the impact force links to the outcomes including production of neurons/astrocytes and thereby boost the novelty of these findings.

A: The specific TBI model we used here shows an increase in neurogenesis, not a reduction. While most studies on TBI agree that neurogenesis is affected, the direction of the change is disputed (see, for example, Clark et al., *Front. Neurosci.*, 2021). As we discuss, the source of variability in TBI literature has been commonly attributed to variations in experimental parameters (injury depth, angle of impact, impact velocity, etc...) and the use of different TBI models. As the reviewer is not specific about which studies they refer to, it is impossible for us to start to analyze all possible experimental conditions that may have impacted the fate of NSC progeny, including their astroglialogenesis potential.

As we explain in the text, our aim here was to model a moderate form of TBI, as nearly half of the patients with moderate TBI experience some form of long-term cognitive impairment and are more likely to develop depression and neurodegenerative diseases. Therefore, we have deliberately selected a reproducible TBI model in mice and have strictly adhered to the protocol outlined in Siebold et al, *Exp Neurol*, 2018. How this compares to other models, that used different injury parameters, is difficult to predict and, albeit interesting, outside the scope of our current manuscript. The CCI-induced TBI model we implemented here, uses 4 different impact parameters (i.e. angle of the impact, depth of injury, impact velocity and dwell time), each of which can vary over a considerable range. This generates so many possible combinations of these 4 parameters, that adequately testing all of them, as the reviewer seems to suggest, would require multiple additional experiments, taking years of additional work for a largely unpredictable outcome. As such, although interesting, such additional studies are not realistic, and we believe beyond the scope of our current revision. However, we have now included an extended discussion of this point in the text (lines 599-606).

6. It is unclear what the behavioural read out adds as it is not linked in any direct functional way with the astroglialogenesis. In addition, it contradicts a number of studies that show reduced neurogenesis and impaired cognitive performance.

A: We have now addressed this query experimentally, as it is also a relevant point raised by Reviewer 1 (point 4). For this, we used injections of a retroviral vector expressing GFP (RV-GFP). The RV-GFP was injected immediately before the CCI-induced TBI or control craniotomy, and mice were sacrificed 15 days post-injection. We observed a significant increase in the percentage of newborn neurons (% of GFP+ cells that express DCX as a fraction of the total number of GFP+ cells) in the TBI group compared to Control. In addition to their numerical increase, we observed morphological changes in newborn neurons in the TBI group.

Specifically, we observed a significantly higher percentage of the newborn neurons in the TBI group possessing dendritic arbors with two primary dendrites emerging directly from the soma, or a main dendrite bifurcating within the first 10 μm from the soma. Newborn neurons in the TBI group also had a significantly lower number of dendritic spines per distance unit. Indeed, our new results further indicate that TBI-induced changes in newborn neurons correlated with the behavioral changes we see in the Morris water maze, consistent with behavioral abnormalities post-TBI resulting from increased neurogenesis. To some degree this interpretation is influenced by the fact that the impact of neuronal heterogeneity on higher brain functions has been established, while the relevance of astroglial heterogeneity is still poorly understood. However, the layer-specific molecular, morphological, and physiological features of dentate gyrus astrocytes suggest they play a role in neural circuit function (Karpf, et al., *Nat. Neurosci.*, 2022). Therefore, the changes in NSC-derived astroglial numbers and location we describe after TBI, may also impact on neural circuit function. This

is an intriguing hypothesis that will require extensive experimental demonstration but is beyond the focus of our current study.

Finally, in the redrafted text we address the link between neurogenesis and cognitive performance as previously described in the literature (see also point 5 of this reviewer). The new data is now presented in the results (lines 135-149); Fig. 1s-v and its amended legend (lines 1113-1124) methods (lines 679-748). An amended discussion has been included to deal with these new data (lines 405-409 and 521-531).

7. It is notoriously difficult to sort for neurons via FACS, hence many groups have focused on sorting neuronal nuclei. The authors appear to have a very good recovery - have other groups published using this approach with similar success?

A: Indeed, as we stated in the text, the expression of GFP in Nestin-GFP mice has been used before in single cell RNA sequencing (scRNA-seq) studies of the AHN niche, to sort NSCs, progenitor and other cell populations, including immature neurons, using flow cytometry (Shin et al., Cell Stem Cell, 2015; Artegiani et al., Cell Rep, 2017) included as references in the revised text. Similarly, other studies have used comparable strategies in which specific NSC populations are labelled with fluorescent proteins controlled by other cell type-specific promoters (Harris et al., Cell Stem Cell, 2021; Hochgerner et al., Nat Neurosci, 2018). This point has now been highlighted in the methods (lines 850-852).

Editorial requests

In addition, we answered the following editorial requests:

POLICIES AND FORMS REQUIRED FOR RESUBMISSION

** Please complete or update the following checklist(s) to verify compliance with our research ethics and data reporting standards. Address all points on the checklist, revising your manuscript in response to the points if needed.*

The form(s) must be downloaded and completed in Adobe Reader rather than opened in a web browser. Each form must be uploaded as a Related Manuscript file at the time of resubmission.

Editorial policy checklist:

<https://www.nature.com/documents/nr-editorial-policy-checklist.pdf>

A: completed

Reporting summary:

A: completed

** Nature journals have recently announced an update to our guidance on reporting on sex and gender in research studies (see here). We strongly encourage researchers to follow the 'Sex and Gender Equity in Research – SAGER – guidelines' and to include sex and gender considerations for studies involving humans, vertebrate animals and cell lines where relevant to the topic of study (an overview can be found here). Authors should use the terms sex (biological attribute) and gender (shaped by social and cultural circumstances) carefully in order to avoid confusing both terms.*

When preparing your revised manuscript, please be aware of our guidance on Sex and Gender reporting).

Please note that we require that the following recommendations from the guidelines are followed:

1. If the research findings apply to only one sex or gender, that must be indicated in the title and/or abstract.

A: Male mice was added to the abstract

2a. For studies involving vertebrates animal and cell lines- The Reporting Summary should include whether sex was considered in the study design.

A: included

2b. For studies involving human research participants- The Reporting Summary should include whether sex and/or gender was considered in the study design and whether sex and/or gender of participants was determined based on self-report or assigned (and methodology used).

A: not applicable

3. Data should be reported disaggregated for sex and gender where this information has been collected and consent has been obtained for reporting and sharing individual-level data; disaggregated numbers for individual experiments must be provided in the source data as appropriate whereas overall numbers may be provided in the Nature Portfolio Reporting Summary.

A: not applicable

Information on the points above should be included in the revised manuscript and detailed in the cover letter.

In addition, please note that if sex- and gender-based analyses have been performed a priori, results should be reported regardless of positive or negative outcome. We discourage conducting post hoc sex- and gender-based analysis if the study design is insufficient (for example, low sample size) to enable meaningful conclusions.

If no sex- and gender-based analyses have been performed, please indicate the reasons for the lack of these analyses in the Reporting Summary.

** Your paper uses custom code/software. Please complete the following code and software submission checklist and make your code available for reviewer assessment, if you have not already done so. The code/software can be provided in a zip file with a readme.txt file or other instructions for installing and running the software. If appropriate, also provide example data and expected output. If you have any issues with the file upload, please let me know.*

<https://www.nature.com/documents/nr-software-policy.pdf>

A: custom codes, including a readme.txt file, have been publicly available at https://github.com/araboapresyan/rna_velocity_analysis as indicated now in the code availability section

DATA AND CODE AVAILABILITY

** All Nature Communications manuscripts must include a "Data Availability" section after the Methods section but before the References. If any of the data can only be shared on request or are subject to restrictions, please specify the reasons and explain how, when, and by whom the data can be accessed. For more information on this policy and a list of examples, see:*

<https://www.nature.com/documents/nr-data-availability-statements-data-citations.pdf>

A: Data availability section has been added

** Please also include a “Code Availability” section after the “Data Availability” section. If the code can only be shared on request, please specify the reasons. For more information on our code sharing policy and requirements, please see:*

<https://www.nature.com/nature-portfolio/editorial-policies/reporting-standards#availability-of-computer-code>

A: Code availability section has been added

** As Nature Portfolio policies strongly encourage you to share your research data in a public repository (e.g. spreadsheets, text, images), we are partnering with the figshare repository so that you can use the figshare integration via the ‘Research Data Deposition’ tab when submitting your revised manuscript.*

Data are stored privately until a manuscript decision is reached and you can edit/withdraw them up to this point: you retain rights and control over your data. The data will be published at the same time as your article; you will receive a data DOI, with guidance on linking the data and manuscript. In the event your manuscript is not accepted, you can keep or remove your data in figshare.

We recommend the use of discipline-specific repositories where available and for a number of data types this is mandatory. Ensure you do not submit these data types or any sensitive data to figshare.

** We strongly encourage you to deposit all new data associated with the paper in a persistent repository where they can be freely and enduringly accessed. We recommend submitting the data to discipline-specific and community-recognised repositories; a list of repositories is provided here: <http://www.nature.com/sdata/policies/repositories>*

Refer to our data policies here: <https://www.nature.com/nature-portfolio/editorial-policies/reporting-standards#availability-of-data>

** All novel microarray, DNA sequencing, RNA-seq or proteomic datasets must be deposited in a publicly accessible database, and accession codes and associated hyperlinks must be provided in the “Data Availability” section.*

A: Data repositories and persistent links are indicated in the data availability section

** To maximise the reproducibility of research data, we strongly encourage you to provide a file containing the raw data underlying the following types of display items:*

- Any reported means/averages in box plots, bar charts, and tables*
- Dot plots/scatter plots, especially when there are overlapping points*
- Line graphs*

The data should be provided in a single Excel file with data for each figure/table in a separate sheet, or in multiple labelled files within a zipped folder. Name this file or folder ‘Source Data’, and include a brief description in your cover letter. The “Data Availability” section should also include the statement “Source data are provided with this paper.”

To learn more about our motivation behind this policy, please see: <https://www.nature.com/articles/s41467-018-06012-8>

A: a single Excel file with source data has been prepared and is provided with the revised version of the manuscript. The statement “Source data are provided with this paper” has been included in the data availability section.

** We also mandate the presentation of uncropped versions of any gels or blots, labelled with the relevant panel and identifying information such as the antibody used.*

A: not applicable to our work

** Please replace your bar graphs with plots that feature information about the distribution of the underlying data. All data points should be shown for plots with a sample size less than 10. For larger sample sizes, please consider box-and-whisker or violin plots as alternatives. Measures of centrality, dispersion and/or error bars should be plotted and described in the figure legend.*

A: Bar graphs have been replaced by the suggested representations.

ORCID

** Nature Communications is committed to improving transparency in authorship. As part of our efforts in this direction, we are now requesting that all authors identified as ‘corresponding author’ create and link their Open Researcher and Contributor Identifier (ORCID) with their account on the Manuscript Tracking System prior to acceptance. ORCID helps the scientific community achieve unambiguous attribution of all scholarly contributions.*

You can create and link your ORCID from the home page of the Manuscript Tracking System by clicking on ‘Modify my Springer Nature account’ and following these instructions. Please also inform all co-authors that they can add their ORCIDs to their accounts and that they must do so prior to acceptance. For more information please visit <http://www.springernature.com/orcid>

If you experience problems in linking your ORCID, please contact the Platform Support Helpdesk.

A: done

REVIEWERS' COMMENTS

Reviewer #1 (Remarks to the Author):

In the revised version of their manuscript, Bielefelder et al have fully addressed my comments and concerns. The additional experiments on neurogenesis and astrogenesis in vivo as well as the additional explanations are much appreciated and further clarify the paper's statements. I therefore suggest accepting the manuscript for publication in Nature Communications.

Reviewer #2 (Remarks to the Author):

I am pleased to confirm that all my concerns and suggestions have been thoroughly addressed in the revised manuscript. I recommend its acceptance, and I congratulate the authors on their commendable efforts.